# Early life bacteria and sibling exposure associate with restoration of the infant gut microbiome after cesarean section

Jie Jiang[1,2], Casper Sahl Poulsen[1], Ulrika Boulund[1], Shiraz Shah [1], Urvish Trivedi [1,3], Madhumita Bhattacharyya[4], Avidan U. Neumann[4], Darlene L. Y. Dai [5], Charisse Petersen [5], Courtney Hoskinson [5,6], Theo J. Moraes [7], Piushkumar J. Mandhane[8,9], Elinor Simons[10], Meghan B. Azad[10,11], Padmaja Subbarao [7,12,13], Klaus Bønnelykke [1,14], Bo Chawes [1,14], Stuart E. Turvey[5], Søren J. Sørensen [3,16], Jonathan Thorsen [1,14,16] & Jakob Stokholm [1,2,15,16] ✉

Long-term gut microbiome perturbation following Cesarean section (CS) delivery has been associated with an increased risk of developing childhood asthma. Whether such CS-associated microbiome composition can be modulated by environmental exposures or ecological interactions, and thereby mitigate disease risk, is unclear. In the COPSAC$_{2010}$ birth cohort (N = 700), we develop a restoration score quantifying the degree to which the 1-year gut microbiome resembled that of vaginally delivered infants. We identify predictors of this restoration score in the 1-week gut microbiome. In addition, having older siblings is linked to a higher restoration score, mediated by increased abundances of restoration-associated bacteria. The restoration score, including association with delivery mode, older siblings and later asthma as well as early bacterial drivers, is successfully replicated in the independent Canadian birth cohort, CHILD. These insights suggest that specific early-life bacteria and sibling exposure may support microbiome restoration and confer protective effects against asthma risk.

A Cesarean section (CS) can be a life-saving intervention when medically indicated, but can potentially also lead to short-term and long-term health consequences for women and their children[1]. CS is one of the most important factors determining an infant's developing gut microbiome, and in turn, possibly the development and differentiation of their immune system[2]. Such perturbation of the initial infant gut colonization may be involved in the increased risk for diseases in childhood and later in life observed in CS-born individuals[3,4].

The healthy fetus is considered sterile[5] and the newborn child is colonized by the first microbes during and immediately after birth[6–8]. When comparing the gut microbiota composition, infants born by CS have gut microbiomes that resemble those found on the mother's skin surface, while vaginally born infants harbor gut microbiomes more similar to the adult fecal and vaginal microbiome[9,10]. In babies delivered by CS, previous studies have reported disrupted colonization of maternal *Bacteroides* strains, and more prevalent colonization by opportunistic pathogens associated with the hospital environment. A low-*Bacteroides* profile in the infant gut microbiome is considered a hallmark of the microbiome perturbation in CS-born infants[11,12]. However, low-*Bacteroides* profiles can also be found in vaginally delivered infants[13–15], which are usually used as a reference for a healthy infant gut microbiome. Such early microbiomes of vaginally delivered infants, which resembles the CS-perturbed profiles, could be caused by other perturbers such as intrapartum antibiotics[16,17].

**Fig. 1 | Conceptual design of a restoration score. A** illustrates the hypothesis that a CS-perturbed microbiome composition can be influenced by environmental exposures or ecological interactions and be restored towards a vaginal-like microbiome composition. Created in BioRender. Bønnelykke, K. (2026) https://BioRender.com/pjywq7q. **B** shows that the restoration score in this study describes the status of 1-year gut microbiome, a higher restoration score means a child's microbiome resembles being born vaginally, while a low restoration score means that a child's microbiome resembles being born by CS[17] (See also Methods). Created in BioRender. Bønnelykke, K. (2026) https://BioRender.com/pjywq7q. **C** Boxplot of 1-year restoration score according to different groups, where red represents CS delivery, orange represents vaginal delivery with antibiotic treatment, and green vaginal delivery without antibiotic treatment. The box represents the 25th and 75th percentiles, the middle line represents the median, and whiskers extend to the most extreme values within 1.5× interquartile range (IQR). P values were derived from two-sided linear models comparing the restoration scores among these three groups. **D** Bacteria contributing to the 1-year restoration score. Loadings are derived from the sPLS model trained on the COPSAC2010 1-year gut microbiome in the previous work[17]. Negative/Positive loading corresponds to lower/higher genus abundance in vaginally delivered children compared to CS born children.

Microbiota maturation can be described by age-dependent successional stages: a "mature" microbiota contains certain taxa that are common for that child's age group, while an "immature" or delayed microbiota resembles that of a younger child[18]. The microbiome composition of infants matures through three distinct, conserved stages of development, with the genus predominance shifting from *Escherichia* over *Bifidobacterium* to *Bacteroides*[19]. This ecological succession eventually stabilizes during childhood regardless of early perturbations such as birth by CS. However, such early perturbation may have long-lasting functional effects on immune development. Gut microbiota assembly and immune system development are intimately linked in early life[20,21]. The establishment of immune tolerance by introducing microbial antigens happens in the first few weeks of life[22,23]. Delayed maturation caused by early perturbation might result in a loss of tolerance and a pro-inflammatory response, which has been associated with increased risk of food allergy[24] and asthma[25].

The dynamics of the early life gut microbiota has been a key research area[26,27], the modifiability provides a potential to promote health throughout life[28]. Potential approaches include pre- and probiotics[29], vaginal seeding[30,31] and fecal microbiota transplantation (FMT)[32]. Maternal FMT was reported to have the most dramatic effect on microbiota composition, shifting the composition fully to that of the vaginally born infants[8]. However, a CS-perturbed gut microbiota may also naturally recover, a process which may be influenced by environmental and intrinsic ecological factors in the microbiome.

Several environmental factors have been reported to promote maturation of the gut microbiome, including breastfeeding[33], a rural living environment[34], and having older siblings at home[35]. Breastfeeding is recognized as one of the most influential drivers of gut microbiome composition during infancy, which provides a dynamic source of nutrition that delivers live microbes, immunoglobulins, and bioactive compounds essential for shaping the infant gut microbiome[36]. Besides, Infants from rural areas often exhibit a more diverse gut microbiome and early colonization of Bacteroides, which promotes further microbiome maturation[37,38]. Lastly, infants with older siblings tend to have a more mature gut microbiota by the age of one year, and this maturity is linked to a higher diversity of gut bacteria, which potentially mediates the protective effects of siblings in relation to allergies[35,39].

Previously, we described the link between CS and asthma risk which may be partially mediated via a prolonged perturbation of the infant gut microbiome in the 700 children from the Copenhagen Prospective Studies on Asthma in Childhood2010 (COPSAC2010) prospective birth cohort[17]. Only children who retained a CS-like gut microbiome composition at 1 year had increased risk of later asthma independent of the magnitude of their initial perturbation. Conversely, children whose gut microbiota was restored by 1 year, resembling those born vaginally, had a similar, lower, risk of asthma. In this follow-up study, we describe early-life environmental and ecological predictors of such restoration in the 1-year gut microbiome with the aim to enhance our understanding of the natural microbiome restoration process in the context of CS-associated risk of asthma.

## Results

### Characteristics of the COPSAC2010 cohort

Information on maternal and child characteristics were obtained during scheduled visits to the COPSAC clinic. We observed differences in gestational age, parental age, maternal BMI, breastfeeding duration, and hospitalization after birth according to delivery mode (Supplementary Data 1). We derived a 1-year restoration score based on previous work[17], which is found to be negatively associated with asthma risk at 6 years of age. This restoration score is to characterize the restoration of the 1-year gut microbiome - a higher score indicates that a child's 1-year gut microbiome resembles that of vaginally delivered infants (Fig. 1A, B). Compared to vaginal delivery, children born by CS had a significantly lower restoration score (model estimate = −0.32, 95% Confidence Interval (CI) [−0.51,−0.13], $p = 8\text{e-}04$). In the subgroup analysis (Fig. 1C), children born by CS had a lower restoration score compared to vaginal delivery without antibiotics exposure (−0.36[−0.55,−0.16], $p = 3\text{e-}04$). Notably, children born by vaginal delivery whose mother received antibiotics at birth also had a lower restoration score compared to those without antibiotics exposure (−0.23[−0.48, 0.02], $p = 0.07$), and with a score comparable to the CS-born infants (0.13[−0.19,0.45], $p = 0.43$). There were no detectable differences observed between the planned and emergency procedures (0.18[−0.20,0.57], $p = 0.35$).

### Gut microbiome diversity at 1 week associates with 1-year restoration score

Children with higher 1-year restoration scores had a higher gut microbial α-diversity at early time points (Supplementary Data 2). At

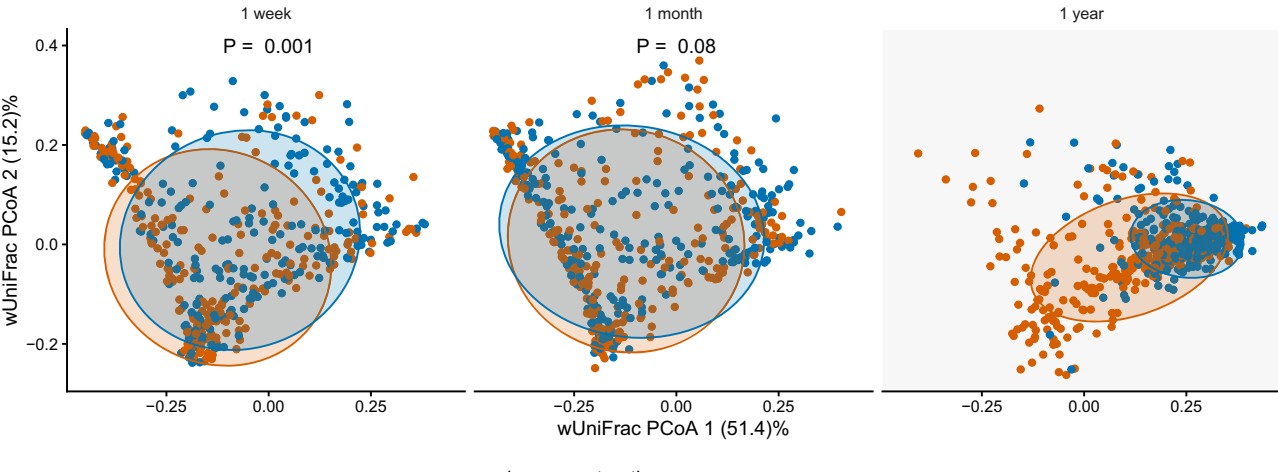

**Fig. 2 | Comparison of the fecal microbiota between children having a low or high 1-year restoration score in the full cohort.** Weighted UniFrac distances were used as input in principal coordinate analysis (PCoA) plots and colored according to children having high restoration scores (above median) (orange, $n = 246, 285, 312$ at 1 week, 1 month and 1 year) and low restoration scores (below median) (blue, $n = 253, 269, 312$ at 1 week, 1 month and 1 year). Gray shading at the 1-year time point marks when the restoration score was derived, hence groups are different by design. PERMANOVA was used to compare group differences, one-sided P values were computed by permutation. Each dot represents one fecal sample. Ellipses represent 1 standard deviation, encompassing approximately 68% of the data points, assuming a bivariate normal distribution.

age 1 week, higher Shannon and Faith's Phylogenetic Diversity (PD) diversity were observed in children with higher 1-year restoration score (Shannon: 0.01[0.001, 0.02], $p = 0.02$, PD: 0.003[0, 0.006], $p = 0.03$). At 1 month of age, the same directionality was observed, but it was not significant for any of the indices. In the CS stratum ($n = 151$), higher PD at 1 week of age was associated (0.006[0,0.01], $p = 0.04$) with higher 1-year restoration score. For other measures, similar estimates were observed but they were not statistically significant (Supplementary Data 2).

The gut microbiota composition differed between children with high and low 1-year restoration scores (above vs below median values) (Fig. 2) using PERMANOVA analysis. From the PCoA plots ordinated across all three timepoints, children with low 1-year restoration scores appeared to have different compositions compared to those with high 1-year restoration scores. The compositional difference of having a high vs low restoration score was most obvious at age 1-week ($F = 4.9$, $R^2 = 0.9\%$, $p = 0.001$) compared to age 1-month ($F = 2.2$, $R^2 = 0.4\%$, $p = 0.08$). In the CS stratum, there were no detectable differences in microbial composition between high and low 1-year restoration score groups at 1 week and 1 month (Supplementary Fig. 1).

### Early gut taxa associate with 1-year restoration score

We next investigated individual taxa at 1 week and 1 month to determine which members of the microbiota were main drivers of the observed compositional differences using the LIMMA model (Fig. 3). At 1 week of age, children with higher restoration scores had higher relative abundances of *Sutterella wadsworthensis* (log fold change (logFC) = 0.12, FDR adjusted p-value ($q$) = 0.03) and *Neglecta timonensis* (logFC = 0.04, $q = 0.03$, Fig. 3A). On the other hand, we found that *Clostridium perfringens* at 1 week was negatively associated with the restoration score and was the most differentially abundant species found (logFC = −0.29, $q = 0.01$), followed by *Enterobacter aerogenes* (logFC = −0.1, $q = 0.03$) and *Actinomyces sp.* (logFC = −0.05, $q = 0.049$). Compared to the gut microbiome at 1 week of age, the 1-month gut microbiome was less associated with the 1-year restoration score. We observed associations between the relative abundances of species at 1 month and restoration scores in the same direction as at 1 week, but they were not FDR significant.

In the CS stratum, higher relative abundances of *Bifidobacterium longum* and *Staphylococcus simiae* at 1 week and only *Streptococcus equinus* at 1 month were associated with higher restoration score at 1 year, but none were FDR significant (Fig. 3B).

### Environmental factors associate with 1-year restoration score

We next examined the relationship between early life environmental factors and the 1-year restoration score (Supplementary Data 3). We found that having older siblings was associated with a higher restoration score in the full cohort (Full cohort, 0.36 [0.21;0.51], $q = 2\text{e-}04$). Interestingly, the age of the youngest older sibling was negatively associated with the restoration score; i.e., the closer in age the older sibling was the higher the restoration score was at 1 year (Full cohort, −0.14 [−0.23;−0.05], $q = 0.04$).

To identify whether specific factors would influence the restoration process differently for children born by CS, we performed this subgroup analysis (Supplementary Data 3). In the CS stratum, having older siblings, having cats, and being born in a rural area were positively associated with the restoration score while being breastfed for more than 6 months was negatively associated with the restoration score, but none were FDR significant.

### Multivariable models on the early life gut microbiome and environmental factors predict the 1-year restoration score

Multivariable analyses were used to identify species and environmental factors jointly associated with the 1-year restoration score in the full cohort (Fig. 4). We employed three sparse Partial Least Squares (sPLS) models on only the 1-week gut microbiome, only environmental factors, and combined 1-week gut microbiome and environmental factors as input features, respectively. These models were evaluated based on the correlations between their cross-validated predictions and 1-year restoration scores (model on only gut microbiome, Spearman rho 0.21, $p = 3\text{e-}06$; model on only environmental factors, 0.16, $p = 7\text{e-}04$; model on combined gut microbiome and environmental factors, 0.24, $p = 2\text{e-}07$). The gut microbiome at 1 week was enriched for *Bifidobacterium longum*, *Parabacteroides distasonis*, *Sutterella wadsworthensis*, and *Neglecta timonensis* in children with higher restoration scores at 1 year, while *Clostridium* perfringens was depleted. Of these, *Bifidobacterium*

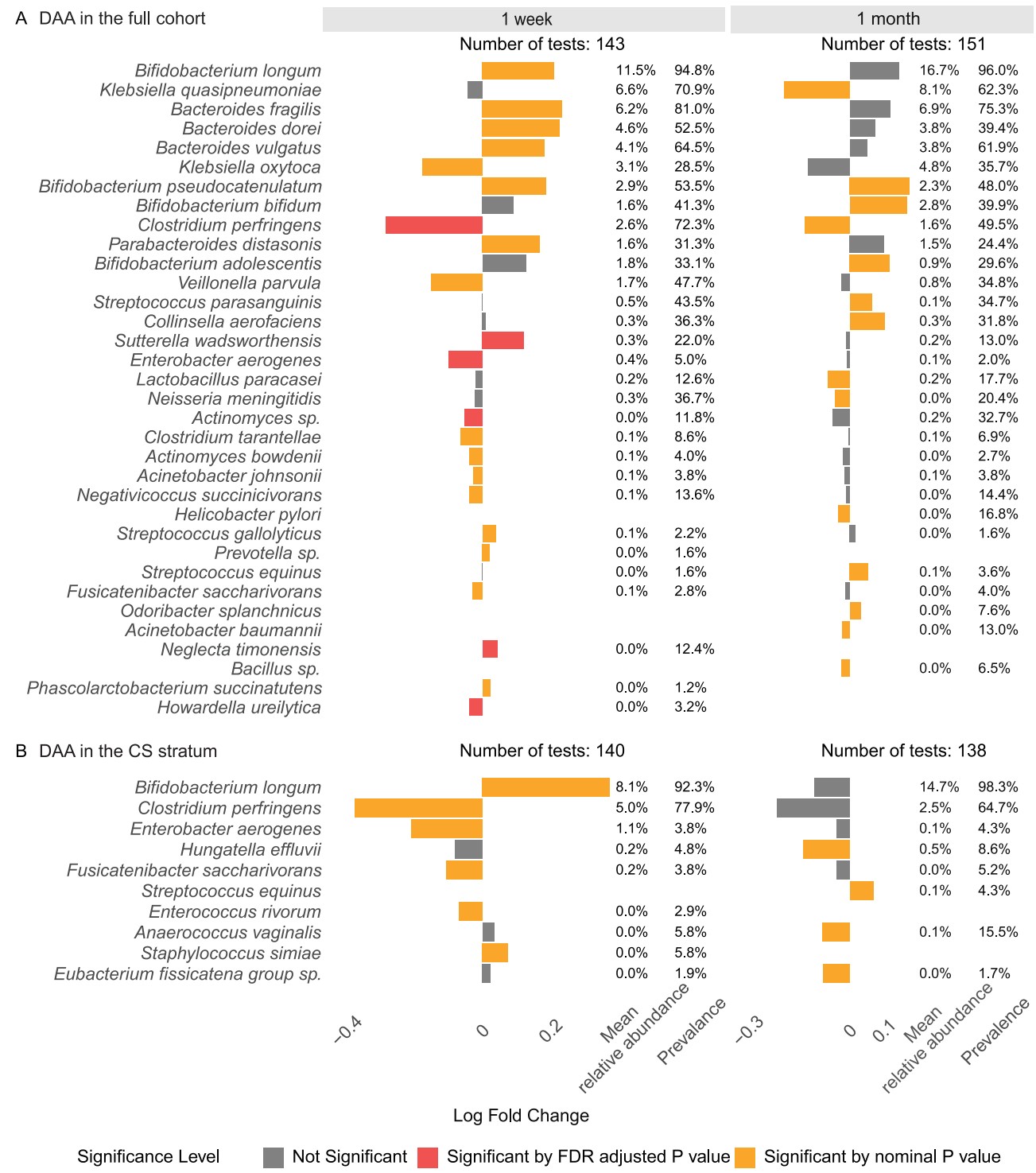

**Fig. 3 | Differential abundance analysis on the species associated with 1-year restoration score.** In the full cohort (**A**) and in the CS stratum (**B**). Species with prevalence of at least 0.1% and relative abundance of more than 0.01% of the total were eligible. The analysis in the full cohort was adjusted for delivery mode. P values were calculated using two-sided tests. The species represented by the red bars were significant after FDR-adjustment, while species represented by yellow bars were nominally significant, and species represented by gray bars were not significant. A positive logFC value indicates an increase in the abundance of the species as the 1-year restoration score increases, while a negative logFC indicates a decrease.

*longum* was the most abundant and prevalent (see Fig. 3), while *Parabacteroides distasonis* and *Sutterella wadsworthensis* were less so. *Neglecta timonensis* was only present in few samples (prevalence 12.4%), but had a strong association with the 1-year restoration score. *Clostridium perfringens* contributed the most to the model with a strong negative loading towards the restoration score. In the model with environmental factors, only maternal antibiotics at birth and

having older siblings had negative and positive loadings with the 1-year restoration score, respectively. The model on combined 1-week gut microbiome and environmental factors showed a better performance than each individual model, predicting the 1-year restoration score with 7 variables selected, including the same five species selected by the model on only the gut microbiome, and *Veillonella parvula* and having older siblings.

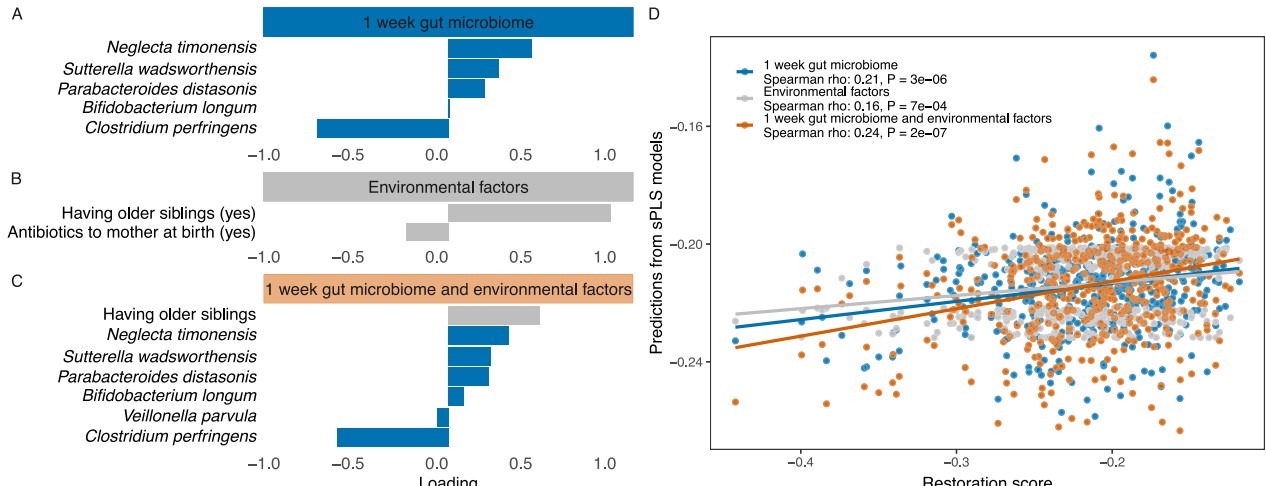

**Fig. 4 | Sparse partial least squares (sPLS) models on restoration score using the full cohort at 1 week of age.** A–C show loadings from models on gut microbiome at 1 week of age (143 species, 499 samples), environmental factors (39 factors, 466 samples), and combined factors (182 variables, 466 samples), respectively. Loadings represent the contribution of each variable to the sPLS models. Negative loadings indicate associations with lower restoration scores, while positive loadings indicate associations with higher restoration scores. **D** Spearman correlation between cross-validated predictions from the above three sPLS models and the 1-year restoration score. The correlation coefficient and two-sided P value are shown in the legend. Blue for the sPLS model on gut microbiome, gray for the sPLS model on environmental factors and orange for the sPLS model on combined gut microbiome and environmental factors.

At 1 month of age, the models had moderate performance on predicting restoration scores (Supplementary Fig. 2, model on gut microbiome, Spearman rho 0.10, $p = 0.023$; model on environmental factors, 0.15, $p = 5e-04$; model on combined gut microbiome and environmental factors, 0.16, $p = 4e-04$). The combined model's cross-validated predictions had a higher correlation with 1-year restoration scores than each individual model, but the inclusion of the gut microbiome didn't improve the model much compared to the model on only environmental factors. In this combined model, 3 environmental factors and 14 species were selected (Supplementary Fig. 2). Of all the selected variables, having older siblings and *Clostridium perfringens* were also selected in the 1-week combined model with the same directionality.

In the CS stratum, the comparatively best model was on the combined 1 week gut microbiome and environmental factors, the Spearman rho between the cross-validated predictions and the restorations scores was 0.19, $p = 0.064$ (Supplementary Fig. 3). However, the models on the 1 month gut microbiome and environmental factors in the CS stratum were not able to predict the restoration scores, with Spearman rho correlations around 0. The overall better performance of models on combined gut microbiome and environmental factors suggested that both might contribute to the restoration process of the perturbed gut microbiome.

### Having older siblings facilitates CS restoration through gut microbiome development

We further investigated whether the species associated with having siblings were also associated with the 1-year restoration score, by comparing the logFC of the species abundances between the two outcomes (Fig. 3, Fig. 5, Supplementary Data 4).

At 1 week, 85 species showed a similar logFC association for both having older siblings and the 1-year restoration score (Fig. 5A, 85 points, gray and blue, were in the first and third quadrants at 1 week). Among them, 5 were significantly associated with both outcomes (Fig. 5B). There was a higher abundance of *Bifidobacterium longum* and *Bifidobacterium pseudocatenulatum* in the children with siblings and a high restoration score; and lower abundance of *Actinomyces bowdenii*, *Clostridium tarantellae*, and *Clostridium perfringens* in the children without siblings and with a low restoration score. At 1 month of age, 4 species were significantly associated with both outcomes in the same direction. At 1 year of age, the time point where the restoration score was defined, 63 species showed significant consistent positive or negative association with both outcomes. To determine whether the number of taxa that were significant for both predictors was greater than expected by chance, and whether the direction of their associations was concordant by chance, we calculated empirical p-values based on permutation tests. Both of the empirical p-values were below 0.05, indicating that the overlap and concordant direction are highly unlikely by chance.

In the CS stratum, 87 species, and 73 species showed consistent associations with having older siblings and the restoration score, at 1 week and 1 month time point, respectively, but less significant compared to the associations in the full cohort (Supplementary Fig. 4, Supplementary Data 5). Nevertheless, higher abundance of *Bifidobacterium longum* was still found positively associated with having older siblings and higher restoration score. We also found the same directionality at 1 year of age in the CS stratum, which further suggests a positive role of having older siblings on the restoration of the gut microbiome by such very early influences. The number of taxa associated with both predictors and concordant direction in the CS stratum was significant in the permutation test.

Having identified older siblings as a potential protective factor that may facilitate the restoration of a CS-perturbed gut microbiome, we next considered these factors together: We performed a mediation analysis to investigate the possible mediating role of the 1-week gut microbiome between having older siblings and the restoration of the gut microbiome by 1 year. Here, we used the cross-validated predictions from the sPLS model of the 1-week gut microbiome to predict the 1 year restoration score and expressed it as a 1-week microbial score representing the degree to which a child's 1-week microbiome looked like it would be restored by 1 year of age (Supplementary Fig. 5). After adjusting for delivery, the results showed a significant mediation effect of older siblings on the 1-year restoration score through the 1-week microbial score (model estimate = 0.002[0, 0.005], $p = 0.004$), but also a strong direct effect of older siblings on the 1-year restoration score (model estimate = 0.02[0.009, 0.03], $p < 2e-16$). The indirect pathway accounted for a portion of 11.2% of the total effect (model estimate = 0.11[0.04,0.26], $p = 0.004$). This suggests that older siblings may contribute to the restoration partially through influencing the very early gut microbiome.

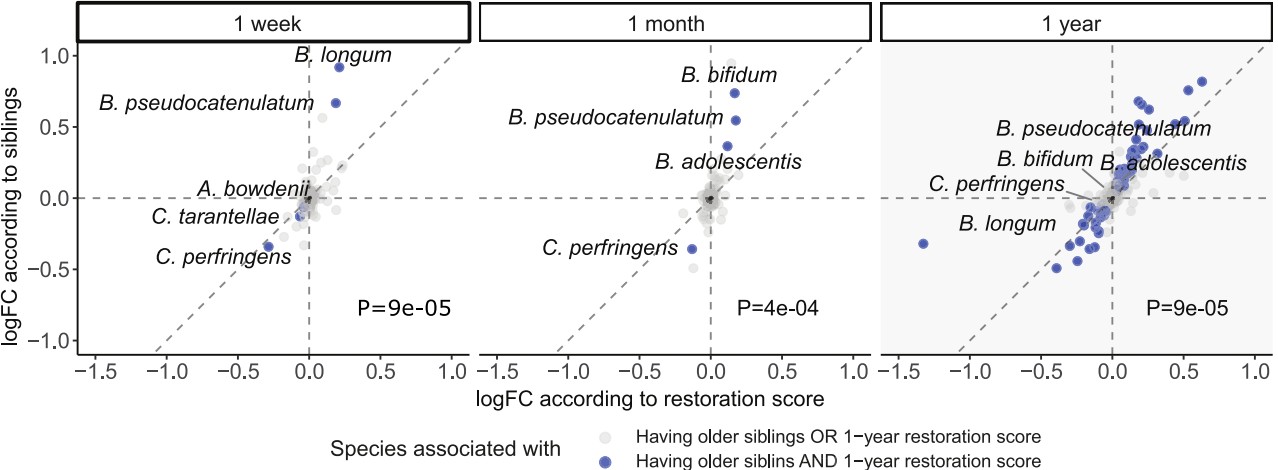

A  Species associated with having older siblings and 1-year restoration score

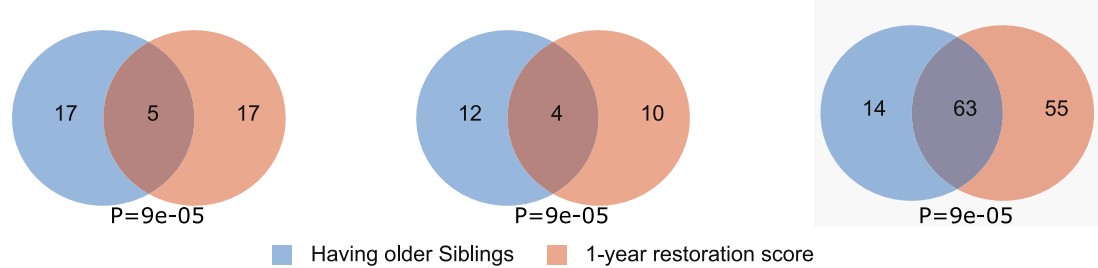

B  Number of species associated with having older siblins and 1-year restoration score

**Fig. 5 | Differential abundance analysis reveals subset of species associated with having older siblings and restoration in the full cohort. A** Scatter plot comparing differential abundance analyses of having older siblings and the restoration score. Each point represents a species, those associated with both factors are highlighted in blue. Labeled species were significant at 1 week and 1 month. A positive logFC value in the first quadrant (upper right) indicates an increase in the abundance of certain bacteria associated with a higher 1-year restoration score and having older siblings at home, while a negative logFC in the third quadrant (bottom left) indicates a decrease in the abundance of certain bacteria associated with a lower 1-year restoration score and not having older siblings. Permutation test P values (upper-tail) assess whether the concordance in direction between the two sets of associations is greater than expected by chance, a P value smaller than 0.05 indicates the concordant direction is unlikely under the null hypothesis. **B** Venn diagram indicating the number of differentially abundant species according to having older siblings and according to the 1-year restoration score. Permutation test P values (upper-tail) assess whether the observed overlap exceeds that expected by chance, a P value smaller than 0.05 indicates the overlap is unlikely under the null hypothesis.

## Early gut taxa and environmental factors associated with restoration scores in the vaginal stratum

While the restoration score was conceived in the context of CS delivery, we also wanted to investigate its dynamics in vaginally delivered children, where low restoration scores would indicate that other factors than CS may have contributed to having a CS-like composition at 1 year. We separately investigated the early gut microbiome associated with the 1-year restoration scores in the vaginal stratum (Supplementary Fig. 6), and perinatal events during vaginal delivery in addition to the previously described prenatal and postnatal factors (Supplementary Data 7). At 1 week of age, children with higher 1-year restoration scores had higher relative abundances of *Sutterella wadsworthensis* (logFC = 0.16, $q = 0.02$) and *Neglecta timonensis* (logFC = 0.06, $q = 0.02$) as we saw in the full cohort. In contrast, *Veillonella parvula* (logFC = −0.22, $q = 0.01$), *Actinomyces sp.* (logFC = −0.08, $q = 0.006$), and *Howardella ureilytica* (logFC = −0.05, $q = 0.01$) were associated with lower restoration scores at 1 year of age in the vaginal stratum. Having older siblings was significantly associated with higher 1-year restoration scores as we saw in the CS stratum (0.02[0.01,0.03], $q = 2e$-03). In contrast, we found antibiotics exposure to mothers and children at birth and meconium-stained amniotic fluid exposure to be negatively associated with the 1-year restoration score, however not FDR significant.

## Validation of the 1-year restoration score on the CHILD cohort

We next sought to validate our findings in an independent dataset. The CHILD study is the largest prospective longitudinal birth cohort study in Canada[40]. We observed similar differences in gestational age, parental age, maternal BMI, and hospitalization after birth according to delivery mode in the CHILD study. On the other hand, compared to the COPSAC cohort, CS-born children had higher antibiotics exposure at 3 months and shorter breastfeeding duration in the CHILD study (Supplementary Data 1).

We applied the sPLS model trained on the COPSAC[2010] cohort[17] to the CHILD cohort's 1-year samples ($n = 325$) to create a restoration score (Fig. 6A). This 1-year restoration score was, like in COPSAC[2010], negatively associated with delivery by CS and positively associated with older siblings in the CHILD cohort (Fig. 6B). Additionally, the CHILD 1-year restoration score was associated with reduced asthma risk at 5 years in both univariate and adjusted logistic regression models (OR 0.63[0.44, 0.87], $p = 0.005$, aOR 0.47 [0.26, 0.82], $p = 0.0084$, Fig. 6B). This replicates the results in our previous study[17] and is consistent with the hypothesis that an appropriate restoration of the gut microbiota could mitigate the increased asthma risk associated with gut microbial changes due to CS delivery.

To identify species at early time points associated with the 1-year restoration score, we trained an sPLS model on the gut microbiome at

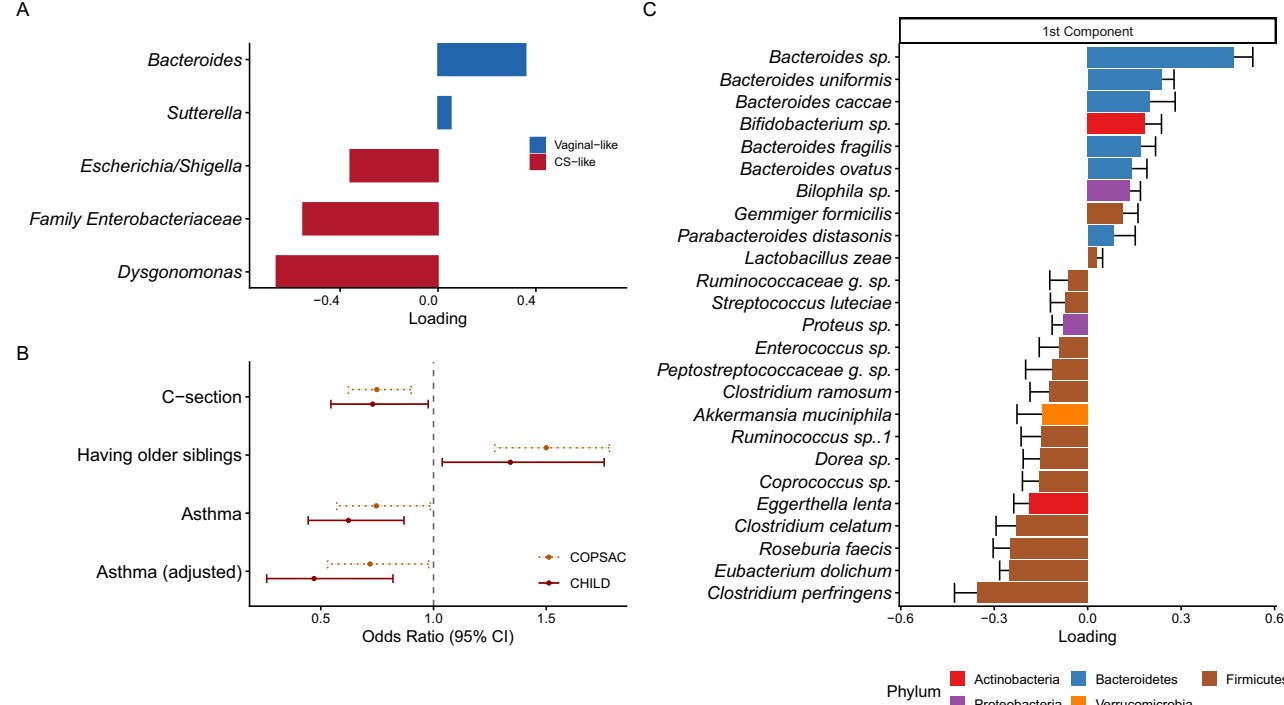

**Fig. 6 | Validation of the COPSAC$_{2010}$ 1-year restoration score in the independent CHILD cohort. A** Loadings are derived from the sPLS model trained on the COPSAC$_{2010}$ 1-year gut microbiome to output a restoration score, the annotations are derived from the CHILD cohort 16 s rRNA data. **B** Univariate logistic regression models on the 1-year restoration score versus CS, having older siblings and asthma, and the adjusted odds ratio from multivariable logistic regression model on asthma, in both cohorts (dashed line: COPSAC; solid line: CHILD). Points represent odds ratios, and error bars represent 95% confidence intervals. The asthma diagnosis was at 5 years in the CHILD study, while the asthma diagnosis was at 6 years of age in the COPSAC cohort. The covariates adjusted in the models were: gestational age, hospitalization after birth, antibiotics exposure to children at 1 year, having older siblings, family asthma history, gender, race, birth season, (and study center for the CHILD study). **C** sPLS model on gut microbiome at 3-month in the CHILD cohort predicting 1-year restoration score, positive loadings indicated positive association with higher restoration score, while negative loadings indicated negative associations with higher restoration score. Bars show mean loadings across repeated cross-validation runs, error bars indicate ±1 standard deviation across $n = 11$ repeated CV runs.

3 months in the CHILD cohort, to predict the 1-year restoration score(Fig. 6C). *Clostridium perfringens* was selected in the model with the highest negative loading as we saw in the COPSAC$_{2010}$ cohort, while *Bacteroides sp.* had the highest positive loading in the model, followed by *Bacteroides uniformis, Bacteroides caccae, Bifidobacterium sp., Bacteroides fragilis,* and *Bacteroides ovatus.* Having older siblings was positively associated with the 1-year restoration score, which supports the idea that older siblings promote the restoration of a CS-perturbed gut microbiome(Fig. 6B).

## Discussion

In this study, we investigated the early-life gut microbiome and environmental factors in relation to restoration of the gut microbiome in the first year of life in a prospective birth cohort. A successful microbiome restoration to a vaginal-born-like composition during this period seems to alleviate the otherwise increased risk of asthma in CS-born children. We found that a higher abundance of several bacteria including members from *Bacteroides* and *Bifidobacterium* was associated with increased restoration of the gut microbiome after CS by 1 year of age. Moreover, having older siblings improved restoration in part mediated via the 1-week gut microbiota. The restoration score, including association with delivery mode, older siblings and later asthma as well as early bacterial drivers, was successfully replicated in the independent CHILD cohort.

During infancy the diet, host genetics, environment and medical interventions determine the establishment and progression of the intestinal microbiota, which interacts with and trains the developing immune system[41]. We found that *Bacteroides* and *Bifidobacterium* species were among the microbiome features at 1 week of age that were associated with a higher restoration score at 1 year. Their abundance drives delivery mode-dependent infant gut microbiota developmental trajectories[15]. By occupying ecological niches within the gut, *Bacteroides* can protect against pathogenic bacteria[42], in part via the production of a unique capsular polysaccharide known as polysaccharide A[43–45].

In the first year of life *Bifidobacterium* has been linked with lower diversity, but its dominance helps establish a protective gut environment[46,47]. However, the timing of having a high *Bifidobacterium* abundance seems to matter. While we observed positive associations with higher abundance early, we also found a negative association between the 1-year *Bifidobacterium longum* abundance and the restoration score (Supplementary Data 5, Supplementary Data 6). Some studies indicate that *Bifidobacterium longum* subsp. *infantis* may colonize the gut later in the breastfeeding period, evidenced by a detectable abundance starting only at 10 weeks of age[48] or later[49]. In addition, the timing of infant colonization with *Bifidobacterium longum* subsp. *infantis* is consistent with horizontal transmission of this subspecies, rather than the vertical transmission previously reported for other *Bifidobacterium* species[50]. In the CS stratum, we observed a negative association between breastfeeding for longer than 6 months with the restoration score, suggesting that breastfeeding until 6 months may decelerate restoration, perhaps through a persistent high *Bifidobacterium* colonization. In a Swedish cohort, *Bifidobacterium longum* subsp. *infantis* reached its peak prevalence at 17 weeks followed by a gradual decline in both prevalence and relative abundance by 52 weeks[48,51]. This dominance of *Bifidobacterium* competes against *Bacteroides* and other bacteria contributing to the restoration,

which could be further associated with a persisting CS-like gut microbiome composition.

We also found other species associated with restoration score, but there is limited research relating them to the restoration. *Neglecta timonensis* is phylogenetically close to *Clostridium sporosphaeroides* and *Clostridium leptum*[52], which are producers of short-chain fatty acids such as butyrate, acetate and propionate from fermentation of fibers, which might play an immunomodulatory role in disease such as asthma[53,54]. *Sutterella wadsworthensis* has been isolated from a variety of clinical specimens throughout the human gastrointestinal tract, in both healthy individuals and patients with gastrointestinal diseases like inflammatory bowel disease (IBD)[55,56]. Few studies describe the role of *Sutterella wadsworthensis*. However, an FMT study showed an enrichment of *Sutterella wadsworthensis* and other species and increased levels of heme and lipopolysaccharide biosynthesis in IBD patients who did not achieve remission after FMT[57]. *Parabacteroides distasonis* has been studied for its potential probiotic properties, with studies suggesting it can produce pentadecnoic acid[58–60].

Conversely, the relative abundance of *Clostridium perfringens* was at all time points negatively associated with the restoration score at 1 year of age in both the full cohort and in the CS stratum, suggesting a negative role in the restoration from a perturbed gut microbiome composition. *Clostridium perfringens* is a common colonizer of the infant gut microbiome, and can colonize the infant very early[61]. While being part of the normal gut microbiota, overgrowth of specific strains can disrupt the microbial diversity. In particular, strains of *Clostridium perfringens* harboring a gene encoding the toxin perfringolysin O (*pfoA*) have been associated with necrotizing enterocolitis (NEC) in preterm infants when present in high abundance[62]. It is reported that CS born infants exhibited higher carriage of toxigenic *Clostridium perfringens*, and infants carrying toxigenic *Clostridium perfringens* had lower levels of *Bacteroides*, *Bifidobacterium*, and *Lactobacillus* groups[63].

We also tested the association between different environmental factors and the 1-year restoration score. Having older siblings has been associated with increased gut microbial diversity during early childhood[39,64]. We have previously reported that having a sibling captures microbial signatures that associate with protection against asthma at 6 years of age, and additionally that presence of siblings promotes a more mature gut microbiome composition at 1 year, which in turn also associates with protection from asthma[65] and allergy[35]. This suggests a long-lasting effect of having older siblings on the gut microbiome and disease risk. In line with a previous study in COPSAC[17], we found a positive association between older siblings and the restoration score, and an association with key gut microbial taxa who appeared as partial mediators of this association, including the enriched *Bifidobacterium longum*, *Bifidobacterium pseudocatenulatum* and *Bifidobacterium adolescentis* and the depleted *Actinomyces bowdenii*, *Clostridium tarantellae*, and *Clostridium perfringens*. Siblings share many strains, which might be due to direct contact between the siblings or due to parallel colonization from a common source, such as the mother or the environment[66]. The transfer of beneficial microorganisms from older siblings could help in the development of the infant's immune system, providing protection against allergic and autoimmune disorders[67]. Furthermore, we found that this protective effect started as early as 1 week of age, and lasted at least till 1 year of age.

In previous studies, breastfeeding contributed to shaping a specific microbial composition in early life, which had the potential to influence infant immune development[33]. We therefore hypothesized that breastfeeding could also promote the restoration of a CS-perturbed gut microbiome. However, we did not observe this association, probably due to the imbalanced groups as the vast majority of children were breastfed at 1 week (exclusive breastfeeding 90%,

$N = 625$) and 1 month (exclusive breastfeeding 79%, $N = 548$) of age, resulting in a small overall variation. Lastly, being born in a rural area and having cats at home were found to be positively associated with the 1-year restoration score only in the CS stratum, which suggests a positive role in restoring a CS-perturbed gut microbiome.

We were able to perform an external validation of our restoration score in an independent cohort, which greatly strengthens the interpretability and robustness of this metric. Here, we applied the previously developed model - trained solely on COPSAC$_{2010}$ 1-year gut microbiome[17] - to the CHILD 1-year gut microbiome. Without any retraining or recalibration, the restoration score produced in the CHILD dataset not only recapitulated the association with delivery mode seen in COPSAC$_{2010}$ cohort, but also demonstrated biological relevance by showing significant association with lower asthma risk at 5 years.

The predictive power of the restoration score in this separate cohort indicates that it can distinguish microbiome development patterns in a biologically consistent way, independent of cohort-specific features and geographical context. This supports our interpretation that the score represents a continuum of microbiota restoration - from perturbed (CS-like) to normalized (vaginal-like). It also affirms that the score captures biologically meaningful variation in vaginally born infants, which we interpret not as a flaw, but as an opportunity to infer the variation in CS born infants, since we included those born by vaginal delivery and received antibiotics at birth in the previous work, and this perturbation could cause a "CS-like" microbiome in vaginally born infants[17]. In the current study, we observed meconium-stained amniotic fluid exposure to be negatively associated with restoration scores in the vaginal stratum.

By showing that infants with higher restoration scores exhibit early microbiome profiles and taxa compositions more typical of vaginally born infants, we believe that these taxa (e.g., *Bacteroides fragilis* and *Bifidobacterium longum*) could contribute to the restoration of a perturbed gut microbiome. On the other hand, we found early *Clostridium perfringens* negatively associated with restoration in both cohorts, which could add on the knowledge of the development of a perturbed gut microbiome. Last but not least, we also validated the positive association between having older siblings and restoration score, which is also consistent with the hygiene hypothesis[68].

There are some limitations to our study. Due to the limitation of 16S rRNA sequencing, our analysis was not able to delineate the composition of subspecies in *Bifidobacterium longum*, which would require metagenomic data or strain isolation. In the beta diversity analyses of the early time points in relation to later restoration score, the variance explained was modest, which is a common observation in microbiome studies due to the high dimensionality and dynamic nature of the infant gut microbiome. This likely reflects the rapid ecological changes occurring during the first year of life, where many taxa are competing in this developing niche. Given the observational design of the study, we only inferred associations between gut microbiome at early and late timepoints and environmental factors. Many of our single ASV results were not significant after FDR correction, which is important for interpretation. We chose to report them here so that others may attempt replication. Furthermore, our analysis approach included multivariable analyses that do not require FDR correction but lack the inference on individual taxa.

The early-life gut microbiome and environmental factors are important for the restoration of a CS-perturbed gut microbiome. In the future, this can be key to further reduce long-term health risks associated with a perturbed gut microbiome such as childhood asthma. Beneficial bacteria like *Bifidobacterium longum* appear to play an important role in restoration, and having older siblings is associated with the restoration process. These findings underscore the potential for early interventions to improve health outcomes in CS-born infants.

## Methods

### The COPSAC$_{2010}$ cohort

**Ethics.** The study was conducted in accordance with the guiding principles of the Declaration of Helsinki and was approved by the Local Ethics Committee (H-B-2008-093) and the Danish Data Protection Agency (2015-41-3696). Both parents gave oral and written informed consent before enrollment.

**Study population.** The COPSAC$_{2010}$ cohort is a population-based mother-child cohort of 700 children and their families, recruited in pregnancy and followed prospectively at the COPSAC research unit. The children were followed by COPSAC study physicians and nurses collecting all biosamples, clinical measurements and diagnoses during clinical visits scheduled at 1 week, 1, 3, 6, 12, 18, 24, 30, and 36 months, thereafter yearly until the age of 6 and again at age 8 and 10 years.

**Study endpoints.** This study is a follow-up of our prior study[17], investigating the gut microbiome development and asthma risk after birth by CS in the COPSAC$_{2010}$ mother-child cohort.

In the present study, we focused on the association between early environmental factors and fecal microbiota composition at 1 week and 1 month, using a 1-year restoration score as the outcome measure. This restoration score was first introduced in our previous paper as a CS score, calculated by constructing a cross-validated sparse partial least squares (sPLS) model on gut microbial composition at 1 year of age predicting delivery mode (vaginal/CS)[17]. The model identified the gut microbial composition at 1 year of age most associated with CS delivery. By reversing the CS microbial score, we define this restoration score as our outcome measure to characterize the restoration of 1-year gut microbiome. Thus, a higher restoration score means a child's microbiome resembles being born vaginally, while a low restoration score means that a child retains a CS-like gut microbiota composition. This score was used as either a continuous score or dichotomized (above and below median value).

**Fecal sample collection and sequencing.** All children with fecal samples collected and characterized by 16S rRNA sequencing (requiring at least 2000 reads) at any of the three time points of 1 week ($n = 552$), 1 month ($n = 607$), and 1 year ($n = 625$) were included in the analyses. Fecal samples were collected either at the research clinic or by the parents at home using detailed instructions. Each sample arrived at the laboratory within 24 h and was mixed on arrival with 1 ml of 10% (v/v) glycerol broth (SSI, Copenhagen, Denmark) and frozen at −80 °C. DNA was extracted using the PowerMag Soil DNA Isolation Kit (MO-BIO Laboratories, Inc., Carlsberg, CA, USA) on an epMotion 5075 (Eppendorf), amplified using a two-step polymerase chain reaction (PCR) with 515 F and 806 R primers flanking the V4 region of 16S rRNA gene, and sequenced using the v2 kit (paired-end 250–base pair reads) on the MiSeq platform (Illumina Inc., San Diego, CA). A full description of the laboratory workflow has been described previously[65].

**Bioinformatic processing.** Raw fastq files were demultiplexed using the MiSeq controller software prior to downstream analysis. As described in previous study[69], the primers and adaptors in sequencing reads were removed using Cutadapt[70]. The determination of amplicon sequence variants (ASVs) was performed on QIIME2 Core 2020.11 platform[71] using Amplicon Denoising Algorithm 2 (DADA2) analysis pipelines[72]. The resulting ASV sequences were annotated using the AnnotIEM[73] pipeline (v.1.3), which combines sequence alignment against four databases: EzBioCloud[74] (r. 2018-05), NCBI[75] (v. refseq 202), RDP[76] (v.11.5), and Silva[77] (v. 138SSU) followed by a high confidence selection of best probable annotation[73]. Genus annotations were correctly annotated for all genera in the mock community, and 15/20 was annotated to the correct species. Therefore, species annotations should be considered putative.

**Maternal and child characteristics.** Information on antibiotics in pregnancy, antibiotics to mother and child at birth, antibiotics to children during the first year of life, gestational age, age when starting daycare, maternal age, maternal BMI, having older siblings at home, age of the youngest sibling, physician-diagnosed asthma, rhinitis, and dermatitis in the mother and father, paternal age, duration of exclusive and total breastfeeding period, hospitalization after birth, any furred animal at home, cats at home, dogs at home, and birth area was obtained during the scheduled visits to the research clinic. Delivery mode was encoded as "vaginal" and "CS" (including planned and emergency CS). Information on intrapartum antibiotics was validated against birth records from hospitals and information on antibiotic use during pregnancy and childhood was validated against national registries[78]. Information on perinatal exposures (asphyxia including meconium-stained amniotic fluid, vacuum-assisted delivery, and pre-labor membrane-rupture) was obtained from national registries[78]. Living environment was based on home address at birth using the CORINE satellite-based land cover database[79].

### The CHILD cohort

**Ethics.** Ethical approval for the CHILD Cohort Study, including the oversight of the CHILD biological samples and the CHILD database (CHILDdb), was obtained from the local Research Ethics Board of each study site: the University of British Columbia, the University of Alberta, the University of Manitoba, the Hospital for Sick Children and McMaster University.

**Study population.** CHILD study is a prospective longitudinal birth cohort study, which enrolled 3405 subjects since pregnancy from 4 largely urban study centers across Canada (Vancouver, Edmonton, Winnipeg, and Toronto) from 2008 to 2012[40]. All children with fecal samples collected and characterized by 16 s rRNA sequencing at around 1 year visit (9 months to 12 months, $n = 325$) were included in the analyses (Supplementary Data 8). Questionnaires related to environmental exposures, psychosocial stresses, nutrition and general health were administered at recruitment, prenatally, at 3, 6, 12, 18, 24, 30 months, and at 3, 4, and 5 years. Covariates used in this study were: delivery mode, gestational age, hospitalization after birth, gender, antibiotics exposure to children at 1 year, family asthma history, race (mother and father), birth season and study center.

**Study endpoints.** Childhood asthma was diagnosed (as Yes/Possible/No) by an expert study physician at the clinical assessment at the age of 5 years based on published approach[40]. For this study, children were considered to have asthma only if the response was 'Yes' and the asthma phenotype was defined as comparing children with asthma at 5 years versus children without asthma at 5 years, children diagnosed as "possible" were excluded.

**Fecal sample collection and sequencing.** Sequencing data generation for infant stool microbiota has been previously described[80]. Briefly, a soiled diaper was provided on the same day for infant stool collection. Samples were refrigerated at home for up to 24 h before being collected and processed by study staff. An additional infant stool sample was provided at the 1-year clinical assessment. DNA was extracted from fecal samples using the commercial kits (Qiagen Mo Bio PowerSoil) optimized for the Thermofisher KingFisher® robot. The V4 hypervariable region of the 16S rRNA gene of fecal DNA was amplified by PCR using universal bacterial primers (V4-515f: V4-806r).

**Bioinformatic processing.** Pooled PCR amplicons were subjected to paired-end sequencing on the Illumina MiSeq platform. Using VSEARCH and Deblur[81] within the QIIME2 pipeline[71], forward and reverse demultiplexed reads were assembled for a final length of 247 bp (unassembled sequences were discarded) and filtered against

the GREENGENES reference database ((v13·8)[82,83]. Taxonomic classification of the resulting unique amplicon sequence variants (ASVs) was achieved using a naïve Bayes classifier trained on reference reads extracted from the reference database at 97% sequence similarity.

**Statistics and data analysis.** Chao 1 index, Shannon diversity and Faith's Phylogenetic Diversity (PD) were used as measures of the within-sample diversity (α-diversity). Chao 1 index and Shannon diversity were calculated using estimate_richness() from package phyloseq v1.48.0, PD was calculated using pd() from package picante v1.8.2. Linear regression was used for analyzing simple associations between α-diversity and the 1-year restoration score. A p-value of 0.05 was considered statistically significant. The between-sample diversity metrics (β-diversity) were computed as weighted UniFrac distances[84]. Differences in β-diversity were visualized with principal coordinates analysis (PCoA) plots, where the restoration score was dichotomized into high and low groups (above/below the median), and tested for inference using permutational multivariable analysis of variance (PERMANOVA; adonis2 from the package vegan v2.6-8 with 999 permutations). A p-value of 0.05 was considered statistically significant.

At species level, we filtered species based on the threshold of mean relative abundance >0.01%. Differential abundance analysis against 1-year restoration score at species level was performed using LIMMA[85] (Linear Models for Microarray Data, DA.lli2 from DAtest[86] package), which fits a linear model to compositional data (log-transformed) for each taxa, multiple testing was controlled using the False Discovery Rate (FDR) adjustment[87] within the function. The associations between the 1-year restoration score and environmental factors were tested using linear regression, multiple testing was controlled using the FDR adjustment. We performed supervised sparse partial least squares (sPLS) regression models on fecal microbiome and environmental factors separately and jointly using package mixOmics v6.28.0. This multivariable model is designed to reduce the high dimensionality of data and perform simultaneous variable selection. By using sPLS, we were to reveal microbial taxa and environmental factors most descriptive of 1-year restoration score. We log$_{10}$-transformed counts data, using the lowest non-zero value as a pseudocount, and then calibrated the data for sequence depth by adjusting the number of reads and the log-transformed number of reads in the linear model. We selected the optimum number of input variables using 11-repeated 10-fold cross-validation of the correlation statistic to avoid overfitting and chose the median repeat for stability. The performance of the model was evaluated by assessing the correlation between predictions from the model and true restoration score. To evaluate the mediation role of the 1-week gut microbiome in the association between having older siblings and the 1-year restoration score, we applied mediation analysis using R package mediation (v4.5.1). A p-value of 0.05 was considered statistically significant. We performed a permutation test to determine whether the number of taxa that are significant for both the 1-year restoration score and older siblings' status was greater than expected by chance, and whether the direction of their associations was concordant by chance. Using 10,000 permutations, we randomly shuffled the two predictor vectors in the data at each time point, recalculated differential abundance for each predictor (adjusting for reads and delivery mode), and merged the results by species. For each permutation, we counted how many species were significant for both predictors and computed the Spearman correlation between their log-fold-changes. The empirical p-values were calculated as P_perm_rho = (sum(rho_perm >= rho_obs) +1) / (10,000 + 1) and P_perm_n = (sum(n_perm >= n_obs) +1) / (10,000 + 1) for the direction and the number of associations, respectively. Analyses were conducted in the full cohort and within the stratum of children born by CS throughout and for vaginal born

children for specific analyses. Regression-based P values were calculated using two-sided tests, permutation-based P values were calculated using one-sided (upper-tail) permutation tests. All analyses were conducted in R v4.4.0.

The validation study was conducted by applying the sPLS model from our previous study[17] to the CHILD 1-year gut microbiome. In the previous work, the sPLS model was trained on 1-year gut microbiome in the COPSAC$_{2010}$ cohort and selected 5 genera to predict if the child was born by CS or not (1/0). Those taxa were: *Bacteroides*, Family *Enterobacteriaceae*, *Sutterella*, *Escherichia/Shigella*, and *Dysgonomonas*. We found the same or the closest annotation in the CHILD cohort as the input of the model: *Bacteroides*, Family *Enterobacteriaceae*, *Sutterella*, *Escherichia/Shigella*, and *Dysgonomonas*. The preprocessing of the abundance was the same as described in the original work, where we agglomerated the rank into genus level, log transformed the relative abundance, using half the lowest nonzero value as a pseudocount. We then used the predict() function from mixOmics package to apply the model on these five genera, and output the *CHILD 1-year restoration score*. To test the association between the CHILD 1-year restoration score and asthma endpoint, we performed both univariate logistic regression and multivariable logistic regression. The covariates adjusted in the multivariable logistic regression models were: gestational age, hospitalization after birth (days), gender, antibiotics exposure to child at 1 year, maternal asthma history, paternal asthma history, maternal race, paternal race, birth season, and study center. The performance of the model was evaluated by associating the CHILD 1-year restoration score with the outcome of the model - delivery mode - using the univariate logistic regression. And the validation of the association between having older siblings and restoration score was tested using univariate logistic regression as well.

### Reporting summary
Further information on research design is available in the Nature Portfolio Reporting Summary linked to this article.

## Data availability
Individual-level personally identifiable clinical data from the children participating in the cohort cannot be made publicly available, to protect the privacy of the participants and their families, in accordance with the Danish Data Protection Act and European Regulation 2016/679 of the European Parliament and of the Council (GDPR) that prohibit distribution even in pseudo-anonymized form. However, research collaborations are welcome, and data can be made available under a joint research collaboration by contacting the COPSAC Data Protection Officer (DPO), Ulrik Ralfkiaer, PhD (administration@dbac.dk). Requests will be answered within two weeks. Data use is restricted to purposes within childhood health and disease. The informed consent obtained from the CHILD participants, in addition to the CHILD Inter-Institutional Agreement (IIA) which has been executed between the five Canadian institutions responsible for the study, govern the sharing of CHILD data. The accession numbers for the 16S rRNA gene sequence data reported in this paper are BioProject accession (NCBI): PRJNA481046. Data described in the manuscript are available by registration to the CHILD database (https://childstudy.ca/childdb/) and the submission of a formal request. All reasonable requests will be accommodated. More information about data access for the CHILD Cohort Study can be found at https://childstudy.ca/for-researchers/data-access/. Researchers interested in collaborating on a project and accessing CHILD Cohort Study data should contact https://child@mcmaster.ca.

## Code availability
All code used in this study is publicly available at https://github.com/finally-jay/restoration-score.git.

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

## Acknowledgements

We thank the children and families of the COPSAC$_{2010}$ cohort study for support and commitment. We acknowledge and appreciate the efforts of the COPSAC research team. Furthermore, we are grateful to all the families who took part in the CHILD study and the whole CHILD team. The COPSAC team is supported by a variety of private and public research funds, listed on www.copsac.com. The Lundbeck Foundation (R16-A1694), the Ministry of Health (903516), the Danish Council for Strategic Research (0603-00280B) and the Capital Region Research Foundation have provided core support to the COPSAC research centre. J.J., K.T., and J.S. were supported by the European Research Council (grant number 101125482). S.E.T. holds a Tier 1 Canada Research Chair in Pediatric Precision Health and the Aubrey J. Tingle Professor of Pediatric Immunology. D.L.Y.D. is funded by a Canadian Institute of Health Research Frederick Banting and Charles Best Canada Graduate Scholarship Doctoral Award (CIHR CGS-D) and the University of British Columbia Four Year Doctoral Fellowship (4YF). M.B.A. holds a Tier 2 Canada Research Chair in Early Nutrition and the Developmental Origins of Health and Disease and is a Fellow if the CIFAR Humans and the Microbiome Program. P.S. holds a Tier 1 Canada Research Chair in Pediatric Asthma and Lung Health. P.J.M. holds Women's and Children's Health Research Institute. Funding for this specific study came from Genome Canada and Genome British Columbia (grant to S.E.T. [274CHI]) with additional support B.C. Children's Hospital Research Institute and Foundation, as well as the Provincial Health Services Authority.

## Author contributions

Conceptualization: J.S. and J.T. Methodology: J.S., J.T., C.S.P., and J.J. Investigation: J.J. Visualization: J.J., J.S., J.T., and C.S.P. Supervision: J.S., J.T., and C.S.P. Writing—original draft: J.J. Writing—review & editing: J.J., C.S.P., U.B., S.S., U.T., M.B., A.U.N., D.L.Y.D., C.P., C.H., T.J.M., P.J.M., E.S., M.B.A., P.S., K.B., B.C., S.E.T., S.J.S., J.T., and J.S.

## Competing interests

The authors declare no competing interests.

## Additional information

[1]COPSAC, Copenhagen Prospective Studies on Asthma in Childhood, Copenhagen University Hospital - Herlev and Gentofte, Copenhagen, Denmark. [2]Department of Food Science, Faculty of Science, University of Copenhagen, Frederiksberg C, Denmark. [3]Department of Biology, Faculty of Science, University of Copenhagen, Copenhagen, Denmark. [4]Institute of Environmental Medicine and Integrative Health, Faculty of Medicine, University of Augsburg, Augsburg, Germany. [5]Department of Pediatrics, BC Children's Hospital, University of British Columbia, Vancouver, BC, Canada. [6]Department of Microbiology and Immunology, University of British Columbia, Vancouver, British Columbia, Canada. [7]Department of Pediatrics, The Hospital for Sick Children, Toronto, Canada. [8]Department of Pediatrics, University of Alberta, Edmonton, Canada. [9]Department of Medicine, Faculty of Medicine and Health Sciences, UCSI University, Kuala Lumpur, Malaysia. [10]Section of Allergy and Immunology, Department of Pediatrics and Child Health, University of Manitoba, Winnipeg, MB, Canada. [11]Manitoba Interdisciplinary Lactation Centre (MILC), Children's Hospital Research Institute of Manitoba, Winnipeg, MB, Canada. [12]Department of Medicine, McMaster University, Hamilton, ON, Canada. [13]Dalla Lana School of Public Health, University of Toronto, Toronto, Canada. [14]Department of Clinical Medicine, Faculty of Health and Medical Sciences, University of Copenhagen, Copenhagen, Denmark. [15]Department of Pediatrics, Slagelse Hospital, Slagelse, Denmark. [16]These authors jointly supervised this work: Søren J. Sørensen, Jonathan Thorsen, Jakob Stokholm. ✉e-mail: stokholm@copsac.com

