## [Transparent Peer Review file · Nature Communications]

Early life bacteria and sibling exposure associate with restoration of the infant gut microbiome after cesarean section

Corresponding Author: Dr Jakob Stokholm

Version 0:

Reviewer comments:

Reviewer #1

(Remarks to the Author)

Jiang and colleagues describe the impact of specific fecal bacteria present in the first weeks of life and environmental factors on a “restoration score” of the microbiome at 1 year of age upon Cesarean section delivery using data from the COPSAC2010 study. Moreover, these findings were replicated in the Canadian CHILd study.

While most of the analyses are sound and the question on what is driving development of the microbiome in early life is a relevant topic, I have some reservations regarding the interpretation and the framing of the data.

The authors developed a score that captures to what extent the microbiome of C-section delivered infants is restored towards a vaginal-like microbiome composition by 1 year of age. The findings are interesting, particularly with respect to the association between the score and asthma risk which was also replicated. My main concerns is however the framing of this score as a restoration score for the C-section disturbed microbiome for several reasons:

- It suggests that vaginal born infants have a “normal” or undisturbed microbiome (Fig 1A), but in fact the authors show that also within vaginally delivered infants many have a relatively low restoration score at 1 year. In fact, there is a significant overlap between the restoration scores of C-section and vaginally delivered infants as can be seen in Fig1C (both with only slightly different medians but also largely overlapping ranges). Is the restoration score actually statistically significantly different unadjusted or when adjusted for important confounding factors? This information seems lacking from Figure 1. In general conceptually these data indicate that this score captures amongst others delayed colonization with *Bacteroides* species, something that is not only limited to C-section born infants but also to part of the vaginally born infants.

- The explained variance of the early-life microbiota on the restoration score appears very low (R^2 0.9% and 0.4% for the 1 week and 1 month microbiota data). It would be good to elaborate on the magnitude of the impact beyond merely the statistical significance.

- The authors themselves conclude that vaginally born infants could have a CS-like gut microbiome – for example due to antibiotic-induced perturbations. Many vaginally born infants without antibiotic exposure also have a CS-like microbiome suggesting that other determinants, likely intrapartum events (e.g., (premature) rupture of amniotic membranes, defecation during labor, assisted vaginal delivery) could play an important role. It would therefore be of significant added value when the authors could elucidate what explains the CS-like microbiome in vaginally delivered newborns based on amongst other intrapartum events.

- In this respect also the distinction between emergency and elective C-section would likely play an important role – could the authors distinguish between scheduled and emergency procedures?

- Most of the associations that the authors report in relation to the restoration score are only statistically significant in the full cohort and not in the stratum of CS-delivered infants. For example, no detectable differences were observed in microbial composition at 1 week and 1 month between CS-delivered infants in the high- and low 1-year restoration groups. Also, no differentially abundant taxa in the CS stratum were observed upon adjustment of multiple comparisons. Finally none of the environmental determinants appeared to be significantly linked to the restoration score in the CS stratum. This further

suggests that the observed findings are not specific for CS-delivered infants but more likely resemble a different maturation among infants that start off with amongst others a low abundance of Bacteroides and Sutterella, and high abundance of Escherichia-Shigella and other Enterobacteriaceae.

- While, given the focus on restoration of the microbiota composition in CS-born infants, it might seem logical to only present the CS stratum in subgroup analyses from a methodological point of view it is important to also present and discuss the VD stratum. It would be important to check if the few findings that were observed in the CS stratum in association to the restoration score would be similar or not in the VD stratum. If similar findings would be observed in both strata this would further underscore that it is mainly a general restoration of delayed colonization during birth rather than a CS-specific restoration.

Additional comments:

- The phrasing "one genus from the family Enterobacteriaceae" as one of the main loadings in CHILD cohort data (Fig 6) is somewhat ambiguous. It is not entirely clear whether this refers to a single, well-defined genus that could not be further annotated, or whether it represents a catch-all category of sequencing reads within Enterobacteriaceae that could not be classified to the genus level and might therefore comprise a heterogeneous set of taxa. Clarifying this point would help readers better interpret the taxonomic resolution and biological relevance of this finding.

- Shigella comes out as another important loading in the CHILD cohort data but this is extremely unlikely as Shigella is a strict pathogen resulting in dysentery. This is a typical and common misclassification as Shigella is basically pathogenic lineage of E. coli and is often 100% identical to other E. coli lineages at the 16S v4 region. It would be important to correct this in order to not provide wrong suggestions.

Minor comments:

- Line 148 – the Chao1 index is not a microbial diversity index but a richness index

- Fig 2 – The PCoA on the 1-year data is somewhat suggestive and misleading as the strong difference is to be expected since the 1-year microbiota was used to create the restoration score.

- Would be nice to apply permutation testing on the bacteria associated with both having older siblings and 1-year restoration index to check if number of bacteria associated with both is statistically significantly higher than expected based on the associations with either one of these alone.

(Remarks on code availability)

Reviewer #2

(Remarks to the Author)

In this manuscript, the authors showed using sparse Partial Least Squares models several microbiome and environmental factors that contribute to the so-called restoration of the gut microbiome in caesarean-section born children in a previously established cohort. Furthermore, the authors were able to externally validate the models' performance using a separate cohort, also previously established, from a different child population. The results seem to be sufficiently discussed and references relevant to the field of early microbiome development, C-section and asthma have been included both in the introduction and discussion. The methodology and interpretation of results seem sound, although I was left with a couple of comments in these sections that might help the authors improve the manuscript further. Overall, I would recommend sending the manuscript back to the authors for minor revisions.

-Results, overall: Continually, both FDR-adjusted and nominal P-values are being used here. The results get somewhat confusing when these results are being mixed and matched, sometimes both are being reported, and sometimes a P-value has been reported without clarification on whether it's nominal or FDR-adjusted. I think it would be best to either stick to one (preferably FDR-adjusted values) or write the results more clearly in this regard.

-Figure 1D, Figure 4, Figure 6C: What does "loading" in the x-axis mean? Might be helpful to explain this somewhere.

-Results, line 149: Here's an abbreviation that has not been explained earlier (PD). This should be opened up here.

-Results, line 176: The figure reference here (Fig. 2A) does not seem correct. This should be fixed.

-Results, Figure 6B: Does the legend mean that the top line is COPSAC dataset and bottom line is CHILD? Might be useful to also have some visual differences, such as dotting in one or having differently colored lines to highlight which result represents which dataset.

-Methods: The methods don't describe how the statistical testing for the environmental factors associated with 1-year restoration were performed. It would be helpful if this was also included here.

(Remarks on code availability)

While not having physically tested whether the code works, I have confirmed that the code for the manuscript has a working link, includes detailed instructions on performing the analyses and the code itself seems good.

Version 1:

Reviewer comments:

Reviewer #1

(Remarks to the Author)

The authors have significantly revised the manuscript and addressed all remarks I previously raised. Especially the additional analyses on microbial and environmental factors associated with the restoration score in the vaginally born infants are very interesting and of additional value. I have no further comments.

(Remarks on code availability)

Reviewer #2

(Remarks to the Author)

The manuscript has been significantly improved with revisions. I have no further comments.

(Remarks on code availability)

I did suggest including the code for the analysis. I appreciate that the methods have been very much expanded but it seems that the authors have not shared their code. I would not recommend rejection solely based on this but I would strongly recommend for the authors to reconsider publishing their code for this manuscript as well as any other future manuscripts for transparency.

Thank you for the constructive and insightful comments. Concerns about the framing of the restoration scores and the statistical testing are very valid, and we have fully and thoroughly addressed them in this revised version of the manuscript. We also considered the suggestion to perform analyses in the vaginal strata and have incorporated those analyses into the manuscript. Additionally, we appreciate the detailed comments on terminology and plots. Below is a point-by-point response that addresses all comments and outlines the specific changes made, which have greatly improved the manuscript. We thank the reviewers for their valuable feedback and look forward to any further comments on this revised version.

Reviewer #1 (Remarks to the Author):

Jiang and colleagues describe the impact of specific fecal bacteria present in the first weeks of life and environmental factors on a "restoration score" of the microbiome at 1 year of age upon Cesarean section delivery using data from the COPSAC2010 study. Moreover, these findings were replicated in the Canadian CHILD study.

While most of the analyses are sound and the question on what is driving development of the microbiome in early life is a relevant topic, I have some reservations regarding the interpretation and the framing of the data.

The authors developed a score that captures to what extent the microbiome of C-section delivered infants is restored towards a vaginal-like microbiome composition by 1 year of age. The findings are interesting, particularly with respect to the association between the score and asthma risk which was also replicated. **My main concerns is however the framing of this score as a restoration score for the C-section disturbed microbiome for several reasons:**

- It suggests that vaginal born infants have a "normal" or undisturbed microbiome (Fig 1A), but in fact the authors show that also within vaginally delivered infants many have a relatively low restoration score at 1 year. In fact, there is a significant overlap between the restoration scores of C-section and vaginally delivered infants as can be seen in Fig1C (both with only slightly different medians but also largely overlapping ranges). **Is the restoration score actually statistically significantly different unadjusted or when adjusted for important confounding factors?** This information seems lacking from Figure 1. In general conceptually these data indicate that this score captures amongst others delayed colonization with *Bacteroides* species, something that is not only limited to C-section born infants but also to part of the vaginally born infants.

Response 1: Thank you for this very good observation. According to the comment, we have now added the association tests between the restoration score and delivery mode in the crude group (vaginal/C-section), and in the subgroups (Vaginal, no ABX/Vaginal, ABX/C-section) to the figure. C-section is associated with significantly lower restoration scores (Estimate:-0.32[-0.51,-0.13], $P=8e-04$) compared to vaginal delivery; while in the subgroup analyses, compared to Vaginal, no ABX, Vaginal, ABX was negatively associated with restoration scores (-0.23[-0.48, 0.02], $P=0.07$), so was the C-section group with pronounced significance (-0.36[-0.55,-0.16], $P=3e-04$). There was no detectable difference in the restoration scores between the Vaginal, ABX and CS group(-0.13[-0.45,0.19], $P=0.43$), which suggests vaginally born infants whose mother received antibiotics at birth can indeed have a skewed gut microbiome, which is also the point of using the restoration score.

These tests have now been added to the plot Fig. 1C, and Fig 1 legend. Line 139

- The explained variance of the early-life microbiota on the restoration score appears very low (R^2 0.9% and 0.4% for the 1 week and 1 month microbiota data). It would be good to elaborate on the magnitude of the impact beyond merely the statistical significance.

Response 2: Thank you for the comments. The restoration score explains only a modest proportion of the variance in the early gut microbiota as quantified by the adonis2 permanova. This is a common observation in microbiome studies due to the high dimensionality and dynamic nature of the infant gut

microbiome. This likely reflects the rapid ecological changes occurring during the first year of life, where many taxa are competing in this developing niche.

We have now addressed this in the limitation section of the discussion. Line 592

- The authors themselves conclude that vaginally born infants could have a CS-like gut microbiome – for example due to antibiotic-induced perturbations. Many vaginally born infants without antibiotic exposure also have a CS-like microbiome suggesting that other determinants, likely intrapartum events (e.g., (premature) rupture of amniotic membranes, defecation during labor, assisted vaginal delivery) could play an important role. It would therefore be of significant added value when the authors could elucidate what explains the CS-like microbiome in vaginally delivered newborns based on amongst other intrapartum events.

Response 3:

Thank you for these comments. We agree that it is of interest and have now performed additional analyses in the vaginal stratum, including how the restoration score is associated with environmental factors and the early gut microbiome. Please refer to response 6 for details.

- In this respect also the distinction between emergency and elective C-section would likely play an important role – could the authors distinguish between scheduled and emergency procedures?

Response 4: Thank you for bringing up this point. We have tested the difference in the restoration scores between the planned and emergency procedures with linear regression using planned C-section as the reference, and no detectable differences were observed (Estimate 0.18[-0.20,0.57], P=0.35, number of planned / emergency C-section: n = 69 / 93).

For this reason we have kept all C-sections in one category. We have added the additional analysis in the results. Line 146

- Most of the associations that the authors report in relation to the restoration score are only statistically significant in the full cohort and not in the stratum of CS-delivered infants. For example, no detectable differences were observed in microbial composition at 1 week and 1 month between CS-delivered infants in the high- and low 1-year restoration groups. Also, no differentially abundant taxa in the CS stratum were observed upon adjustment of multiple comparisons. Finally none of the environmental determinants appeared to be significantly linked to the restoration score in the CS stratum. This further suggests that the observed findings are not specific for CS-delivered infants but more likely resemble a different maturation among infants that start of with amongst others a low abundance of Bacteroides and Sutterella, and high abundance of Escherichia-Shigella and other Enterobacteriaceae.

Response 5: Please refer to response 6.

- While, given the focus on restoration of the microbiota composition in CS-born infants, it might seem logical to only present the CS stratum in subgroup analyses from a methodological point of view it is important to also present and discuss the VD stratum. It would be important to check if the few findings that were observed in the CS stratum in association to the restoration score would be similar or not in the VD stratum. If similar findings would be observed in both strata this would further underscore that it is mainly a general restoration of delayed colonization during birth rather than a CS-specific restoration.

Response 6:

Thank you for this very relevant suggestion. It is indeed interesting to examine prenatal and postnatal factors among vaginally delivered infants who exhibit a “CS-like” gut microbiome. In this response, we consider not only the restoration score at 1 year of age but also the early restoration scores at 1 week and 1 month (again derived from Stokholm, Science Transl Med 2020). As with the 1-year score, the 1-week and 1-month scores indicate the extent to which an infant’s gut microbiome resembles that of

infants born vaginally. Within the vaginal-delivery stratum, low restoration scores quantify how similar the gut microbiome is to that of infants born by CS, reflecting perturbations due to other factors. These scores were calculated separately, predicting CS on gut microbiome at each time point, but they are highly correlated (restoration score 1 week vs 1 month: ρ 0.52, $P < 2e-16$; 1 week vs 1 year: ρ 0.16, $P = 3e-04$; 1 month vs 1 year: ρ 0.17, $P = 4e-05$).

We examined associations between these restoration scores in the vaginal stratum and various perinatal factors (intrapartum antibiotics, prelabor membrane rupture, and Meconium-Stained Amniotic Fluid (MSAF) exposure) as well as postnatal factors (hospitalization, presence of older siblings, and all other factors previously reported in the manuscript). As you can see from the table below, antibiotics to mothers and children at birth and hospitalization after birth were negatively associated with early restoration scores. MSAF exposure was negatively associated with the 1-year restoration score. There is a limited number of studies investigating the influence of MSAF exposure on the infants' gut microbiome, but it's reported that MSAF exposure can be a risk factor for many different conditions.

We next looked into the bacterial species at 1 week and 1 month associated with restoration scores in the vaginal stratum. At 1 week of age, children with higher restoration scores had higher relative abundances of *Sutterella wadsworthensis* and *Neglecta timonensis* as we saw in the full cohort (FDR adjusted $P < 0.05$). In contrast, *Veillonella parvula*, *Actinomyces sp.*, and *Howardella ureilytica* were associated with lower restoration scores at 1 year of age (FDR adjusted $P < 0.05$).

We acknowledge the relevance of the inclusion of analyses in the vaginal stratum, and have made a whole new separate paragraph in the results section investigating the associations with 1-year restoration scores in the manuscript:

"Early gut taxa and environmental factors associated with restoration scores in the vaginal stratum",
Line 389

Environmental factors Estimate[LowCI,HighCI], P Stats	Restoration score at 1 week	Restoration score at 1 month	Restoration score at 1 year
Gestational age in days	0.06[-0.08,0.19],P=0.42 N=396	0.13[0.01,0.25],P=0.03 N=434	0[0,0.01],P=0.4 N=491
Antibiotics to child at birth	-0.93[-1.53,-0.34],P=2e-03 no 384 / yes 9	-0.73[-1.26,-0.2],P=7e-03 no 419 / yes 12	-0.02[-0.05,0],P=0.05 no 473 / yes 15
Antibiotics to mother at birth	-0.36[-0.63,-0.1],P=7e-03 no 343 / yes 50	-0.26[-0.51,0],P=0.05 no 374 / yes 57	-0.01[-0.02,0],P=0.07 no 425 / yes 63
Preeclampsy	-0.22[-0.67,0.22],P=0.33 no 361 / yes 16	-0.26[-0.72,0.2],P=0.27 no 392 / yes 17	-0.01[-0.03,0.01],P=0.38 no 443 / yes 21
Induction	-0.12[-0.34,0.1],P=0.27 no 291 / yes 85	-0.08[-0.3,0.13],P=0.44 no 306 / yes 102	-0.01[-0.02,0],P=0.17 no 350 / yes 113
Asphyxia	-0.1[-0.34,0.13],P=0.38 no 307 / yes 70	-0.06[-0.29,0.17],P=0.63 no 328 / yes 81	-0.01[-0.02,0],P=0.27 no 371 / yes 93
Meconium-stained amniotic fluid	-0.3[-0.7,0.1],P=0.14 no 357 / yes 20	-0.03[-0.44,0.38],P=0.88 no 387 / yes 22	-0.02[-0.04,0],P=0.02 no 440 / yes 24
Vacuum-assisted delivery	-0.16[-0.46,0.14],P=0.3 no 339 / yes 38	-0.17[-0.48,0.13],P=0.27 no 369 / yes 40	0[-0.02,0.01],P=0.66 no 418 / yes 46
Prelabor membrane-rupture	-0.19[-0.52,0.15],P=0.28 no 347 / yes 30	-0.19[-0.53,0.15],P=0.28 no 377 / yes 32	-0.01[-0.03,0],P=0.1 no 427 / yes 37
Prelabor membrane-rupture >= 37w	-0.21[-0.56,0.14],P=0.25 no 350 / yes 27	-0.15[-0.51,0.21],P=0.42 no 381 / yes 28	-0.01[-0.03,0.01],P=0.25 no 432 / yes 32
Preterm-Prelabor membrane-rupture	0.03[-0.99,1.05],P=0.95 no 374 / yes 3	-0.42[-1.35,0.51],P=0.37 no 405 / yes 4	-0.03[-0.08,0.01],P=0.13 no 459 / yes 5
Hospitalization after birth	-0.81[-1.13,-0.49],P=9e-07 no 365 / yes 31	-0.73[-1.02,-0.45],P=8e-07 no 392 / yes 42	-0.01[-0.02,0.01],P=0.3 no 443 / yes 48

Additional comments:

- The phrasing “one genus from the family Enterobacteriaceae” as one of the main loadings in CHILD cohort data (Fig 6) is somewhat ambiguous. It is not entirely clear whether this refers to a single, well-defined genus that could not be further annotated, or whether it represents a catch-all category of sequencing reads within Enterobacteriaceae that could not be classified to the genus level and might therefore comprise a heterogeneous set of taxa. Clarifying this point would help readers better interpret the taxonomic resolution and biological relevance of this finding.

Response 7: Thank you for pointing this out. In the CHILD cohort, the annotation was *Enterobacteriaceae* at the family level, which refers to a generic-level assignment that could not be resolved to a specific genus. This taxon is therefore a heterogeneous, unclassified clade within the *Enterobacteriaceae* that may comprise several related genera, which was at the same level as what we had in the COPSAC cohort. Therefore, we changed the name of the loading into “*Family Enterobacteriaceae*” in the Fig 6, and clarified it in the methods section. Line 783

- *Shigella* comes out as another important loading in the CHILD cohort data but this is extremely unlikely as *Shigella* is a strict pathogen resulting in dysentery. This is a typical and common misclassification as *Shigella* is basically pathogenic lineage of *E. coli* and is often 100% identical to other *E. coli* lineages at the 16S v4 region. It would be important to correct this in order to not provide wrong suggestions.

Response 8: Thank you for this one as well. We fully agree, and have changed the label into “*Escherichia/Shigella*” in Fig 6, and clarified it in the methods section. Line 786

Minor comments:

- Line 148 – the Chao1 index is not a microbial diversity index but a richness index

Response 9: Thank you for pointing this out. We have changed the wording and will be careful not to mix them. Line 727

- Fig 2 – The PCoA on the 1-year data is somewhat suggestive and misleading as the strong difference is to be expected since the 1-year microbiota was used to create the restoration score.

Response 10: We thank you for raising this point, but argue to keep it to show the development over time. To avoid confusion, we have removed the P value at 1 year from the plot, colored the 1-year time point with a grey shadow, and have now clarified this point additionally in the figure legend. Line 189

- Would be nice to apply permutation testing on the bacteria associated with both having older siblings and 1-year restoration index to check if number of bacteria associated with both is statistically significantly higher than expected based on the associations with either one of these alone.

Response 11: Thank you for the great suggestion for analysis, which will strengthen the argument substantially. We have now performed a permutation test to determine whether the number of taxa that are significant for both the 1-year restoration score and older siblings status is greater than expected by chance, and whether the direction of their associations is concordant by chance. Using 10,000 permutations, we randomly shuffled the two predictor vectors in the data at each time point, recalculated differential abundance for each predictor, and merged the results by species. For each permutation, we counted how many species were significant for both predictors and computed the Spearman correlation between their log-fold-changes. The empirical p-values were calculated as $P_{perm_rho} = (\text{sum}(\rho_{perm} \geq \rho_{obs}) + 1) / (10,000 + 1)$ and $P_{perm_n} = (\text{sum}(n_{perm} \geq n_{obs}) + 1) / (10,000 + 1)$ for the direction and the number of associations, respectively. Both of the empirical p-values were below 0.05, indicating that the overlap and concordant direction are highly unlikely under the null.

These results confirm that a subset of gut taxa is indeed jointly associated with 1-year restoration score and sibling exposure. We have added this in the manuscript. Line 350, Fig 5, Line 364, Line 755

Reviewer #2 (Remarks to the Author):

In this manuscript, the authors showed using sparse Partial Least Squares models several microbiome and environmental factors that contribute to the so-called restoration of the gut microbiome in caesarean-section born children in a previously established cohort. Furthermore, the authors were able to externally validate the models' performance using a separate cohort, also previously established, from a different child population. The results seem to be sufficiently discussed and references relevant to the field of early microbiome development, C-section and asthma have been included both in the introduction and discussion. The methodology and interpretation of results seem sound, although I was left with a couple of comments in these sections that might help the authors improve the manuscript further. Overall, I would recommend sending the manuscript back to the authors for minor revisions.

-Results, overall: Continually, both FDR-adjusted and nominal P-values are being used here. The results get somewhat confusing when these results are being mixed and matched, sometimes both are being reported, and sometimes a P-value has been reported without clarification on whether it's nominal or FDR-adjusted. I think it would be best to either stick to one (preferably FDR-adjusted values) or write the results more clearly in this regard.

Response 12: Thank you for this comment, after consideration, we would like to keep the figures with/without FDR adjustment because of the biological meaning (potential to compare between strata and between timepoints, potential for replication by future studies), but we agree to keep it clean in the text, only write about FDR significant findings, when the multiple testing had been done. Line 208-211, Line 241-252, Line 396-404.

-Figure 1D, Figure 4, Figure 6C: What does "loading" in the x-axis mean? Might be helpful to explain this somewhere.

Response 13: Thank you for raising this point of potential confusion for the reader. We have explained the meaning of loadings in each of the figure legends. Line: 160, Line 296, Line 466.

-Results, line 149: Here's an abbreviation that has not been explained earlier (PD). This should be opened up here.

Response 14: Thank you very much for this comment. We have added the full name of Faith's Phylogenetic Diversity (PD) in Line 168.

-Results, line 176: The figure reference here (Fig. 2A) does not seem correct. This should be fixed.

Response 15: Thank you for spotting this error, we have corrected it as Fig.3A in line 205.

-Results, Figure 6B: Does the legend mean that the top line is COPSAC dataset and bottom line is CHILD? Might be useful to also have some visual differences, such as dotting in one or having differently colored lines to highlight which result represents which dataset.

Response 16: Yes, exactly. We have now used different colors to additionally illustrate the differences.

-Methods: The methods don't describe how the statistical testing for the environmental factors associated with 1-year restoration were performed. It would be helpful if this was also included here.

Response 17: Thank you for spotting this, we have added the description of the statistical analysis for the environmental factors in the methods section. Line 742

Reviewer #2 (Remarks on code availability):

While not having physically tested whether the code works, I have confirmed that the code for the manuscript has a working link, includes detailed instructions on performing the analyses and the code itself seems good.

Response 18: Thank you very much.

Thank you for the constructive and insightful comments. Concerns about the framing of the restoration scores and the statistical testing are very valid, and we have fully and thoroughly addressed them in this revised version of the manuscript. We also considered the suggestion to perform analyses in the vaginal strata and have incorporated those analyses into the manuscript. Additionally, we appreciate the detailed comments on terminology and plots. Below is a point-by-point response that addresses all comments and outlines the specific changes made, which have greatly improved the manuscript. We thank the reviewers for their valuable feedback and look forward to any further comments on this revised version.

Reviewer #1 (Remarks to the Author):

Jiang and colleagues describe the impact of specific fecal bacteria present in the first weeks of life and environmental factors on a "restoration score" of the microbiome at 1 year of age upon Cesarean section delivery using data from the COPSAC2010 study. Moreover, these findings were replicated in the Canadian CHILD study.

While most of the analyses are sound and the question on what is driving development of the microbiome in early life is a relevant topic, I have some reservations regarding the interpretation and the framing of the data.

The authors developed a score that captures to what extent the microbiome of C-section delivered infants is restored towards a vaginal-like microbiome composition by 1 year of age. The findings are interesting, particularly with respect to the association between the score and asthma risk which was also replicated. **My main concerns is however the framing of this score as a restoration score for the C-section disturbed microbiome for several reasons:**

- It suggests that vaginal born infants have a "normal" or undisturbed microbiome (Fig 1A), but in fact the authors show that also within vaginally delivered infants many have a relatively low restoration score at 1 year. In fact, there is a significant overlap between the restoration scores of C-section and vaginally delivered infants as can be seen in Fig1C (both with only slightly different medians but also largely overlapping ranges). **Is the restoration score actually statistically significantly different unadjusted or when adjusted for important confounding factors?** This information seems lacking from Figure 1. In general conceptually these data indicate that this score captures amongst others delayed colonization with *Bacteroides* species, something that is not only limited to C-section born infants but also to part of the vaginally born infants.

Response 1: Thank you for this very good observation. According to the comment, we have now added the association tests between the restoration score and delivery mode in the crude group (vaginal/C-section), and in the subgroups (Vaginal, no ABX/Vaginal, ABX/C-section) to the figure. C-section is associated with significantly lower restoration scores (Estimate:-0.32[-0.51,-0.13], $P=8e-04$) compared to vaginal delivery; while in the subgroup analyses, compared to Vaginal, no ABX, Vaginal, ABX was negatively associated with restoration scores (-0.23[-0.48, 0.02], $P=0.07$), so was the C-section group with pronounced significance (-0.36[-0.55,-0.16], $P=3e-04$). There was no detectable difference in the restoration scores between the Vaginal, ABX and CS group(-0.13[-0.45,0.19], $P=0.43$), which suggests vaginally born infants whose mother received antibiotics at birth can indeed have a skewed gut microbiome, which is also the point of using the restoration score.

These tests have now been added to the plot Fig. 1C, and Fig 1 legend. Line 139

- The explained variance of the early-life microbiota on the restoration score appears very low (R^2 0.9% and 0.4% for the 1 week and 1 month microbiota data). It would be good to elaborate on the magnitude of the impact beyond merely the statistical significance.

Response 2: Thank you for the comments. The restoration score explains only a modest proportion of the variance in the early gut microbiota as quantified by the adonis2 permanova. This is a common observation in microbiome studies due to the high dimensionality and dynamic nature of the infant gut

microbiome. This likely reflects the rapid ecological changes occurring during the first year of life, where many taxa are competing in this developing niche.

We have now addressed this in the limitation section of the discussion. Line 592

- The authors themselves conclude that vaginally born infants could have a CS-like gut microbiome – for example due to antibiotic-induced perturbations. Many vaginally born infants without antibiotic exposure also have a CS-like microbiome suggesting that other determinants, likely intrapartum events (e.g., (premature) rupture of amniotic membranes, defecation during labor, assisted vaginal delivery) could play an important role. It would therefore be of significant added value when the authors could elucidate what explains the CS-like microbiome in vaginally delivered newborns based on amongst other intrapartum events.

Response 3:

Thank you for these comments. We agree that it is of interest and have now performed additional analyses in the vaginal stratum, including how the restoration score is associated with environmental factors and the early gut microbiome. Please refer to response 6 for details.

- In this respect also the distinction between emergency and elective C-section would likely play an important role – could the authors distinguish between scheduled and emergency procedures?

Response 4: Thank you for bringing up this point. We have tested the difference in the restoration scores between the planned and emergency procedures with linear regression using planned C-section as the reference, and no detectable differences were observed (Estimate 0.18[-0.20,0.57], P=0.35, number of planned / emergency C-section: n = 69 / 93).

For this reason we have kept all C-sections in one category. We have added the additional analysis in the results. Line 146

- Most of the associations that the authors report in relation to the restoration score are only statistically significant in the full cohort and not in the stratum of CS-delivered infants. For example, no detectable differences were observed in microbial composition at 1 week and 1 month between CS-delivered infants in the high- and low 1-year restoration groups. Also, no differentially abundant taxa in the CS stratum were observed upon adjustment of multiple comparisons. Finally none of the environmental determinants appeared to be significantly linked to the restoration score in the CS stratum. This further suggests that the observed findings are not specific for CS-delivered infants but more likely resemble a different maturation among infants that start off with amongst others a low abundance of Bacteroides and Sutterella, and high abundance of Escherichia-Shigella and other Enterobacteriaceae.

Response 5: Please refer to response 6.

- While, given the focus on restoration of the microbiota composition in CS-born infants, it might seem logical to only present the CS stratum in subgroup analyses from a methodological point of view it is important to also present and discuss the VD stratum. It would be important to check if the few findings that were observed in the CS stratum in association to the restoration score would be similar or not in the VD stratum. If similar findings would be observed in both strata this would further underscore that it is mainly a general restoration of delayed colonization during birth rather than a CS-specific restoration.

Response 6:

Thank you for this very relevant suggestion. It is indeed interesting to examine prenatal and postnatal factors among vaginally delivered infants who exhibit a “CS-like” gut microbiome. In this response, we consider not only the restoration score at 1 year of age but also the early restoration scores at 1 week and 1 month (again derived from Stockholm, Science Transl Med 2020). As with the 1-year score, the 1-week and 1-month scores indicate the extent to which an infant’s gut microbiome resembles that of

infants born vaginally. Within the vaginal-delivery stratum, low restoration scores quantify how similar the gut microbiome is to that of infants born by CS, reflecting perturbations due to other factors. These scores were calculated separately, predicting CS on gut microbiome at each time point, but they are highly correlated (restoration score 1 week vs 1 month: ρ 0.52, $P < 2e-16$; 1 week vs 1 year: ρ 0.16, $P = 3e-04$; 1 month vs 1 year: ρ 0.17, $P = 4e-05$).

We examined associations between these restoration scores in the vaginal stratum and various perinatal factors (intrapartum antibiotics, prelabor membrane rupture, and Meconium-Stained Amniotic Fluid (MSAF) exposure) as well as postnatal factors (hospitalization, presence of older siblings, and all other factors previously reported in the manuscript). As you can see from the table below, antibiotics to mothers and children at birth and hospitalization after birth were negatively associated with early restoration scores. MSAF exposure was negatively associated with the 1-year restoration score. There is a limited number of studies investigating the influence of MSAF exposure on the infants' gut microbiome, but it's reported that MSAF exposure can be a risk factor for many different conditions.

We next looked into the bacterial species at 1 week and 1 month associated with restoration scores in the vaginal stratum. At 1 week of age, children with higher restoration scores had higher relative abundances of *Sutterella wadsworthensis* and *Neglecta timonensis* as we saw in the full cohort (FDR adjusted $P < 0.05$). In contrast, *Veillonella parvula*, *Actinomyces* sp., and *Howardella ureilytica* were associated with lower restoration scores at 1 year of age (FDR adjusted $P < 0.05$).

We acknowledge the relevance of the inclusion of analyses in the vaginal stratum, and have made a whole new separate paragraph in the results section investigating the associations with 1-year restoration scores in the manuscript:

"Early gut taxa and environmental factors associated with restoration scores in the vaginal stratum",
Line 389

Environmental factors Estimate[LowCI,HighCI], P Stats	Restoration score at 1 week	Restoration score at 1 month	Restoration score at 1 year
Gestational age in days	0.06[-0.08,0.19],P=0.42 N=396	0.13[0.01,0.25],P=0.03 N=434	0[0,0.01],P=0.4 N=491
Antibiotics to child at birth	-0.93[-1.53,-0.34],P=2e-03 no 384 / yes 9	-0.73[-1.26,-0.2],P=7e-03 no 419 / yes 12	-0.02[-0.05,0],P=0.05 no 473 / yes 15
Antibiotics to mother at birth	-0.36[-0.63,-0.1],P=7e-03 no 343 / yes 50	-0.26[-0.51,0],P=0.05 no 374 / yes 57	-0.01[-0.02,0],P=0.07 no 425 / yes 63
Preeclampsy	-0.22[-0.67,0.22],P=0.33 no 361 / yes 16	-0.26[-0.72,0.2],P=0.27 no 392 / yes 17	-0.01[-0.03,0.01],P=0.38 no 443 / yes 21
Induction	-0.12[-0.34,0.1],P=0.27 no 291 / yes 85	-0.08[-0.3,0.13],P=0.44 no 306 / yes 102	-0.01[-0.02,0],P=0.17 no 350 / yes 113
Asphyxia	-0.1[-0.34,0.13],P=0.38 no 307 / yes 70	-0.06[-0.29,0.17],P=0.63 no 328 / yes 81	-0.01[-0.02,0],P=0.27 no 371 / yes 93
Meconium-stained amniotic fluid	-0.3[-0.7,0.1],P=0.14 no 357 / yes 20	-0.03[-0.44,0.38],P=0.88 no 387 / yes 22	-0.02[-0.04,0],P=0.02 no 440 / yes 24
Vacuum-assisted delivery	-0.16[-0.46,0.14],P=0.3 no 339 / yes 38	-0.17[-0.48,0.13],P=0.27 no 369 / yes 40	0[-0.02,0.01],P=0.66 no 418 / yes 46
Prelabor membrane-rupture	-0.19[-0.52,0.15],P=0.28 no 347 / yes 30	-0.19[-0.53,0.15],P=0.28 no 377 / yes 32	-0.01[-0.03,0],P=0.1 no 427 / yes 37
Prelabor membrane-rupture >= 37w	-0.21[-0.56,0.14],P=0.25 no 350 / yes 27	-0.15[-0.51,0.21],P=0.42 no 381 / yes 28	-0.01[-0.03,0.01],P=0.25 no 432 / yes 32
Preterm-Prelabor membrane-rupture	0.03[-0.99,1.05],P=0.95 no 374 / yes 3	-0.42[-1.35,0.51],P=0.37 no 405 / yes 4	-0.03[-0.08,0.01],P=0.13 no 459 / yes 5
Hospitalization after birth	-0.81[-1.13,-0.49],P=9e-07 no 365 / yes 31	-0.73[-1.02,-0.45],P=8e-07 no 392 / yes 42	-0.01[-0.02,0.01],P=0.3 no 443 / yes 48

Additional comments:

- The phrasing “one genus from the family Enterobacteriaceae” as one of the main loadings in CHILD cohort data (Fig 6) is somewhat ambiguous. It is not entirely clear whether this refers to a single, well-defined genus that could not be further annotated, or whether it represents a catch-all category of sequencing reads within Enterobacteriaceae that could not be classified to the genus level and might therefore comprise a heterogeneous set of taxa. Clarifying this point would help readers better interpret the taxonomic resolution and biological relevance of this finding.

Response 7: Thank you for pointing this out. In the CHILD cohort, the annotation was *Enterobacteriaceae* at the family level, which refers to a generic-level assignment that could not be resolved to a specific genus. This taxon is therefore a heterogeneous, unclassified clade within the *Enterobacteriaceae* that may comprise several related genera, which was at the same level as what we had in the COPSAC cohort. Therefore, we changed the name of the loading into “*Family Enterobacteriaceae*” in the Fig 6, and clarified it in the methods section. Line 783

- *Shigella* comes out as another important loading in the CHILD cohort data but this is extremely unlikely as *Shigella* is a strict pathogen resulting in dysentery. This is a typical and common misclassification as *Shigella* is basically pathogenic lineage of *E. coli* and is often 100% identical to other *E. coli* lineages at the 16S v4 region. It would be important to correct this in order to not provide wrong suggestions.

Response 8: Thank you for this one as well. We fully agree, and have changed the label into “*Escherichia/Shigella*” in Fig 6, and clarified it in the methods section. Line 786

Minor comments:

- Line 148 – the Chao1 index is not a microbial diversity index but a richness index

Response 9: Thank you for pointing this out. We have changed the wording and will be careful not to mix them. Line 727

- Fig 2 – The PCoA on the 1-year data is somewhat suggestive and misleading as the strong difference is to be expected since the 1-year microbiota was used to create the restoration score.

Response 10: We thank you for raising this point, but argue to keep it to show the development over time. To avoid confusion, we have removed the P value at 1 year from the plot, colored the 1-year time point with a grey shadow, and have now clarified this point additionally in the figure legend. Line 189

- Would be nice to apply permutation testing on the bacteria associated with both having older siblings and 1-year restoration index to check if number of bacteria associated with both is statistically significantly higher than expected based on the associations with either one of these alone.

Response 11: Thank you for the great suggestion for analysis, which will strengthen the argument substantially. We have now performed a permutation test to determine whether the number of taxa that are significant for both the 1-year restoration score and older siblings status is greater than expected by chance, and whether the direction of their associations is concordant by chance. Using 10,000 permutations, we randomly shuffled the two predictor vectors in the data at each time point, recalculated differential abundance for each predictor, and merged the results by species. For each permutation, we counted how many species were significant for both predictors and computed the Spearman correlation between their log-fold-changes. The empirical p-values were calculated as $P_{perm_rho} = (\text{sum}(\rho_{perm} \geq \rho_{obs}) + 1) / (10,000 + 1)$ and $P_{perm_n} = (\text{sum}(n_{perm} \geq n_{obs}) + 1) / (10,000 + 1)$ for the direction and the number of associations, respectively. Both of the empirical p-values were below 0.05, indicating that the overlap and concordant direction are highly unlikely under the null.

These results confirm that a subset of gut taxa is indeed jointly associated with 1-year restoration score and sibling exposure. We have added this in the manuscript. Line 350, Fig 5, Line 364, Line 755

Reviewer #2 (Remarks to the Author):

In this manuscript, the authors showed using sparse Partial Least Squares models several microbiome and environmental factors that contribute to the so-called restoration of the gut microbiome in caesarean-section born children in a previously established cohort. Furthermore, the authors were able to externally validate the models' performance using a separate cohort, also previously established, from a different child population. The results seem to be sufficiently discussed and references relevant to the field of early microbiome development, C-section and asthma have been included both in the introduction and discussion. The methodology and interpretation of results seem sound, although I was left with a couple of comments in these sections that might help the authors improve the manuscript further. Overall, I would recommend sending the manuscript back to the authors for minor revisions.

-Results, overall: Continually, both FDR-adjusted and nominal P-values are being used here. The results get somewhat confusing when these results are being mixed and matched, sometimes both are being reported, and sometimes a P-value has been reported without clarification on whether it's nominal or FDR-adjusted. I think it would be best to either stick to one (preferably FDR-adjusted values) or write the results more clearly in this regard.

Response 12: Thank you for this comment, after consideration, we would like to keep the figures with/without FDR adjustment because of the biological meaning (potential to compare between strata and between timepoints, potential for replication by future studies), but we agree to keep it clean in the text, only write about FDR significant findings, when the multiple testing had been done. Line 208-211, Line 241-252, Line 396-404.

-Figure 1D, Figure 4, Figure 6C: What does "loading" in the x-axis mean? Might be helpful to explain this somewhere.

Response 13: Thank you for raising this point of potential confusion for the reader. We have explained the meaning of loadings in each of the figure legends. Line: 160, Line 296, Line 466.

-Results, line 149: Here's an abbreviation that has not been explained earlier (PD). This should be opened up here.

Response 14: Thank you very much for this comment. We have added the full name of Faith's Phylogenetic Diversity (PD) in Line 168.

-Results, line 176: The figure reference here (Fig. 2A) does not seem correct. This should be fixed.

Response 15: Thank you for spotting this error, we have corrected it as Fig.3A in line 205.

-Results, Figure 6B: Does the legend mean that the top line is COPSAC dataset and bottom line is CHILD? Might be useful to also have some visual differences, such as dotting in one or having differently colored lines to highlight which result represents which dataset.

Response 16: Yes, exactly. We have now used different colors to additionally illustrate the differences.

-Methods: The methods don't describe how the statistical testing for the environmental factors associated with 1-year restoration were performed. It would be helpful if this was also included here.

Response 17: Thank you for spotting this, we have added the description of the statistical analysis for the environmental factors in the methods section. Line 742

Reviewer #2 (Remarks on code availability):

While not having physically tested whether the code works, I have confirmed that the code for the manuscript has a working link, includes detailed instructions on performing the analyses and the code itself seems good.

Response 18: Thank you very much.